# SuperRL: Reinforcement Learning with Supervision to Boost Language Model Reasoning

## Abstract

Large language models (LLMs) are increasingly applied to complex reasoning tasks, where rich high-quality offline data—such as expert-annotated solutions—are often available. However, standard reinforcement learning (RL) struggles in sparse-reward settings and fails to fully leverage offline supervision. We propose SuperRL, a unified training framework that adaptively switches between RL and supervised fine-tuning (SFT). For any data instance where all sampled trajectories yield zero reward, indicating a lack of gradient signal, SuperRL triggers a fallback to supervised fine-tuning using high-quality offline data. Experiments across diverse reasoning benchmarks demonstrate that SuperRL outperforms standard RL, delivering improved sample efficiency, generalization, and robustness.

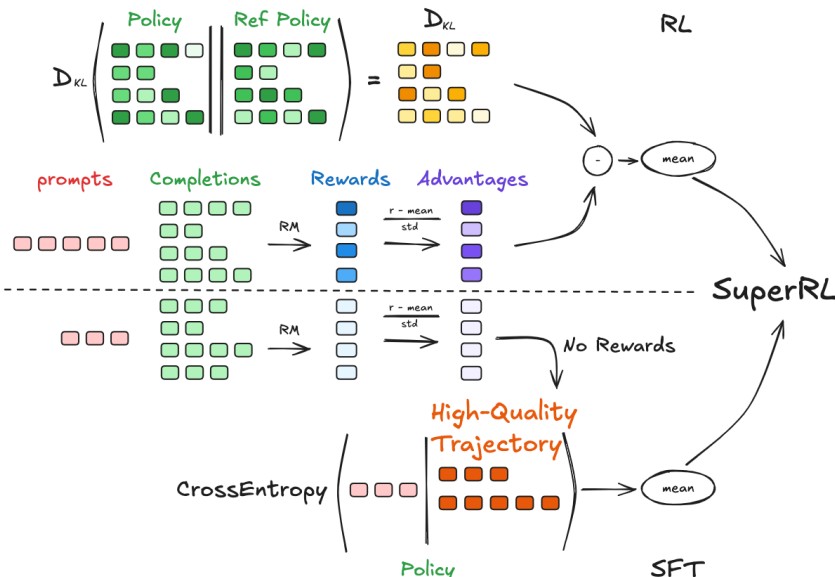

Figure 1: Overview of SuperRL. SuperRL is a unified training framework that adaptively combines RL and SFT based on reward signal. During training, for each input, the model samples multiple rollouts and computes their rewards. If at least one trajectory receives a nonzero reward, standard RL updates are applied using policy gradients. If all trajectories yield zero reward—indicating an absence of learning signal—SuperRL falls back to supervised fine-tuning using high-quality demonstrations.

## 1 Introduction

Large Language Models (LLMs) have revolutionized artificial intelligence across a wide range of domains, from mathematical reasoning and code generation to complex agentic tasks that require structured decision-making. Their ability to adapt to real-world scenarios and deliver task-aligned outputs depends heavily on post-training paradigms—among which Supervised Fine-Tuning (SFT)

and Reinforcement Learning (RL) are the most critical training methods (Lai et al., 2025; Guo et al., 2025). These two approaches play complementary roles in refining LLM capabilities, yet each carries inherent limitations that have long hindered the full realization of their potential when used in isolation or through rigid integration. (Lai et al., 2025; Chu et al., 2025; Ma et al., 2025)

Supervised Fine-Tuning (SFT) excels at distilling knowledge from expert demonstrations and equips Large Language Models (LLMs) with structured problem-solving strategies via static datasets, but it heavily relies on expert data quality and quantity, has limited generalization, suffers from exposure bias, and risks overfitting when extended. (Dong et al., 2023; Chu et al., 2025; Jin et al., 2025) In contrast, Reinforcement Learning (RL) enables LLMs to actively explore the solution space through on-policy learning and refine behaviors via dynamic feedback—enhancing generalization and mitigating SFT's exposure bias—yet it faces inefficient exploration, policy degradation with sparse/noisy rewards, and may fail for weaker models or highly complex tasks. (Chu et al., 2025; Chen et al., 2025a; Shao et al., 2024)

To address these limitations, prior work has proposed integrating SFT and RL, but existing strategies fall short of adapting to the dynamic needs of training. The sequential "SFT-then-RL" paradigm, for example, uses SFT to initialize the model with expert patterns before transitioning to RL for exploration.(Chen et al., 2025a; Shao et al., 2024) While intuitive, this rigid phase separation often triggers a "shift-readapt-overfit" progression: SFT disrupts the model's pre-established reasoning patterns, forcing a costly readaptation during early RL training, and residual overfitting to expert data restricts later exploration.(Zhang et al., 2025) Static mixed-objective approaches, which combine SFT and RL losses with fixed weights or pre-defined schedules, fare no better—they lack the flexibility to respond to real-time training signals. A fixed weight cannot address scenarios where RL provides no meaningful learning (e.g., all rollouts yield zero reward), leading to ineffective exploration, nor can it reduce SFT's influence when RL feedback is abundant, which would otherwise stifle generalization.(Chen et al., 2025c;b; Ma et al., 2025; Yan et al., 2025)

The core insight driving our work is that the optimal balance between SFT and RL depends not on pre-defined rules, but on the availability of learning signals during training. When RL generates non-zero rewards, it provides actionable feedback to refine policies through exploration; when all RL rollouts yield zero reward, the absence of a learning signal renders RL ineffective, and SFT becomes critical to provide expert guidance and avoid stagnation. This observation motivates the design of SuperRL, a unified training framework that adaptively combines SFT and RL based on real-time reward signals.

**Our main contributions are as follows:**

- **Theoretical Justification.** We provide rigorous theoretical analysis for the joint optimization objective of SFT and RL, explicitly proving that SuperRL's reward-driven switching mechanism delivers performance gains. This analysis addresses the inefficiencies of static integration methods—such as RL stagnation in reward-scarce scenarios or residual SFT overfitting—and verifies that adaptive paradigm switching guides the optimization toward a more optimal policy space.

- **SuperRL Framework & Variants.** We develop a general SuperRL framework as the foundation, and further design variants that mainly differ in their formulation of mixed losses. These variants embody distinct philosophies of how SFT and RL should be combined, ranging from stability-oriented to exploration-driven designs. By comparing these alternative loss functions, we are able to examine trade-offs in stability, efficiency, and generalization, and provide flexible options for adapting SuperRL across diverse task settings.

- **Comprehensive Empirical Validation.** We conduct extensive experiments across multiple reasoning benchmarks and model scales. Results demonstrate that SuperRL consistently outperforms standard RL baselines, delivering improvements in sample efficiency, generalization to out-of-distribution problems, and training stability. Beyond empirical performance, our analysis also provides insights into the mechanisms underlying these gains, such as how structured exploration and adaptive credit assignment contribute to more robust reasoning behavior.

## 2 RELATED WORK

### 2.1 THE PITFALLS OF STANDALONE SFT AND RL

Despite their widespread adoption, both SFT and RL exhibit critical limitations when applied in isolation. Pure SFT, while effective at aligning models with expert-labeled data, suffers from over-fitting to annotation biases and limited generalization capacity. Since it lacks mechanisms for ex-ploration, SFT-trained models often struggle with compositional generalization and reasoning under distribution shifts, as observed in the limited performance gains of instruction-tuned models like FLAN-T5 when applied to complex reasoning benchmarks (Chung et al., 2022; Chen et al., 2025a; Chu et al., 2025). Conversely, pure RL introduces its own challenges: it is notoriously sample-inefficient, requiring extensive human preference data or synthetic feedback signals, as seen in early RLHF pipelines for InstructGPT (Ouyang et al., 2022a). Moreover, RL training is often unstable, and when reward signals are sparse or unreliable, models may fail to learn meaningful strategies, a problem documented in subsequent studies of preference optimization and alignment benchmarks (Rafailov et al., 2023; Mukobi et al., 2023; Liao et al., 2025; Rybakov et al., 2024; Arumugam & Griffiths, 2025; Chen et al., 2025a; Chu et al., 2025). Taken together, these limitations highlight that SFT and RL each address complementary aspects of model training—SFT provides strong priors from supervised data, while RL injects adaptivity and exploration—suggesting that their integration offers a more balanced and robust pathway toward building LLMs capable of reliable reasoning and generalization across diverse tasks.(Lv et al., 2025; Zhang et al., 2025; Chen et al., 2025c)

### 2.2 THE PITFALLS OF EXISTING INTEGRATION STRATEGIES

**Limitations of the SFT-then-RL Paradigm.** The most widely adopted integration strategy follows a sequential SFT-then-RL pipeline (Ouyang et al., 2022b; Guo et al., 2025). While intuitive, this approach suffers from an inherent boundary problem: deciding when and how to transition from imitation to exploration. Insufficient SFT leaves the model with disrupted yet unstable reasoning patterns, forcing RL to waste effort on re-adaptation rather than exploration. On the flip side, too much SFT makes the policy too rigid around the expert demonstrations. This takes away the flexi-bility that RL needs. The pattern of "shift-readapt-overfit" phenomenon in SFT training shows that the real problem isn't the loss function itself. It's the switching boundary that's the main bottleneck in the sequential design. (Zhang et al., 2025).

**Alternative Integration Strategies.** To address this boundary challenge, several methods attempt more direct integration of SFT and RL. LUFFY (Yan et al., 2025) augments RL with off-policy reasoning traces, mixing them with on-policy rollouts through a mixed-policy GRPO objective and importance-weighted policy shaping, thereby dynamically balancing imitation and exploration. SRFT (Fu et al., 2025) instead adopts a single-stage framework where demonstrations and self-rollouts are trained jointly, with an entropy-aware weighting that adjusts the trade-off between SFT supervision and RL optimization. ReLIFT (Ma et al., 2025) follows a dynamic scheduling strat-egy, primarily training with RL while injecting targeted SFT updates on the hardest problems to improve efficiency with limited demonstrations. While these approaches mitigate abrupt phase transitions, they still operate at the loss or scheduling level and thus stop short of rethinking the reward–supervision interface itself, leaving open space for integration directly at the reward signal level.

### 2.3 MOTIVATING SUPERRL: ADAPTIVE INTEGRATION GUIDED BY REWARD SIGNALS

The core insight driving our work is this: the optimal balance between SFT and RL depends not on pre-defined rules, but on the availability of reward signals during RL training. When RL generates non-zero rewards, it provides actionable feedback to refine policies through exploration; when all RL rollouts yield zero reward, the absence of a learning signal renders RL ineffective, and SFT becomes critical to provide expert guidance and avoid stagnation.

## 3 METHODOLOGY

### 3.1 THEORETICAL JUSTIFICATION: WHY SFT HELPS WHEN RL HAS NO SIGNAL

A broad class of recent approaches to integrating SFT and RL can be understood through a unified optimization lens. In particular, Lv et al. (2025) propose that both supervised imitation objectives and reinforcement-driven exploration can be captured within a single policy optimization framework:

$$\mathcal{J}_\mu(\theta) = \mathbb{E}_{\tau \sim \pi_\theta}[r(\tau \mid q)] - \mu \cdot \mathrm{KL}(\pi_\beta \,\|\, \pi_\theta),$$

where $\pi_\theta$ is the learned policy, $\pi_\beta$ an expert (or baseline) policy, $r(\tau \mid q)$ the return for trajectory $\tau$ under query $q$, and $\mu > 0$ the SFT weight. Building on this formulation, we provide a theoretical analysis explaining why SFT becomes indispensable in regimes where RL signals vanish.

**Advantage-free regime.** Introduce the advantage of a trajectory relative to the baseline:

$$A(\tau \mid \theta, \beta) = Q(\tau) - V(\pi_\beta),$$

so that $Q(\tau) = V(\pi_\beta) + A(\tau \mid \theta, \beta)$. When the advantage vanishes for all trajectories, $A(\tau) = 0 \,\forall \tau$, we have $Q(\tau) = V(\pi_\beta)$ and therefore

$$\mathbb{E}_{\tau \sim \pi_\theta}[r(\tau \mid q)] = \mathbb{E}_{\tau \sim \pi_\theta}[Q(\tau)] = \mathbb{E}[V(\pi_\beta)],$$

which is constant w.r.t. $\theta$. The objective reduces to

$$\mathcal{J}_\mu(\theta) = \mathbb{E}[V(\pi_\beta)] - \mu \cdot \mathrm{KL}(\pi_\beta \,\|\, \pi_\theta).$$

Since the first term is constant, maximizing $\mathcal{J}_\mu$ is equivalent to minimizing $\mathrm{KL}(\pi_\beta \,\|\, \pi_\theta)$. Concretely, the gradient of the KL term (w.r.t. $\theta$) is

$$\nabla_\theta \mathrm{KL}(\pi_\beta \,\|\, \pi_\theta) = -\mathbb{E}_{\tau \sim \pi_\beta}[\nabla_\theta \log \pi_\theta(\tau)].$$

Hence the gradient of the objective becomes

$$\nabla_\theta \mathcal{J}_\mu(\theta) = \mu \, \mathbb{E}_{\tau \sim \pi_\beta}[\nabla_\theta \log \pi_\theta(\tau)],$$

which is exactly the supervised (behavioral cloning / SFT) update direction. Thus when advantage information disappears (no differential RL signal), SFT provides the only non-trivial direction for policy improvement.

**Zero-reward as an extreme case.** If every trajectory yields zero reward, i.e.

$$r(\tau \mid q) = 0 \quad \text{for all } \tau,$$

then the expected return term vanishes:

$$\mathbb{E}_{\tau \sim \pi_\theta}[r(\tau \mid q)] = 0,$$

and the objective simplifies to

$$\mathcal{J}_\mu(\theta) = -\mu \cdot \mathrm{KL}(\pi_\beta \,\|\, \pi_\theta).$$

Maximizing $\mathcal{J}_\mu$ is therefore equivalent to minimizing the KL divergence; the RL policy gradient contribution is identically zero:

$$\nabla_\theta \, \mathbb{E}_{\tau \sim \pi_\theta}[r(\tau \mid q)] = \mathbb{E}_{\tau \sim \pi_\theta}[\nabla_\theta \log \pi_\theta(\tau) \cdot r(\tau \mid q)] = 0.$$

All learning signal comes from the KL term (i.e., SFT).

**Summary and interpretation.** The two derivations above show the same qualitative point at two granularities:

- If the *advantage* is zero for all trajectories then the RL return term is constant and the optimization reduces to minimizing $\mathrm{KL}(\pi_\beta \| \pi_\theta)$.
- If the *return* itself is zero for all trajectories then the RL gradient is exactly zero and SFT is the only source of gradient.

Therefore, SFT is not merely a convenient warm-start; it serves as an indispensable fallback whenever reinforcement signals fail to produce a useful optimization gradient. This provides a theoretical foundation for our SuperRL design, which adaptively triggers the SFT term depending on whether reward or advantage signals are present, ensuring effective optimization in regions where RL signals are sparse or absent.

## 3.2 SUPERRL DESIGN: ADAPTIVE SFT BASED ON RL SIGNALS

Building on the theoretical insights from the previous subsection, SuperRL employs an adaptive supervised fine-tuning (SFT) mechanism to ensure effective learning in sparse- or zero-signal regimes.

**Adaptive SFT Triggering.** SuperRL dynamically adjusts the contribution of the SFT term based on the informativeness of RL signals on a per-data-instance basis:

- For any data instance where all sampled trajectories yield zero reward (or zero advantage), the SFT term is fully activated, providing the policy with meaningful gradient information.

- For instances with non-zero reward or advantage, the SFT term can be scaled down or omitted, allowing RL-driven exploration and optimization to dominate.

---

**Algorithm 1** SuperRL Algorithm

---

**Require:** data instance $q$, current policy $\pi_\theta$, reference policy $\pi_\beta$
**Ensure:** updated policy $\pi_\theta$ with adaptive SFT
1: Sample trajectories $\{\tau_i\} \sim \pi_\theta(\cdot \mid q)$
2: Compute reward $r(\tau_i \mid q)$ or advantage $A(\tau_i \mid q)$ for each trajectory
3: **if** all $r(\tau_i \mid q) = 0$ **or** $A(\tau_i \mid q) = 0$ **then**
4:     Activate SFT: $\mu(q) \leftarrow 1$
5: **else**
6:     Scale down or skip SFT: $\mu(q) \leftarrow 0$
7: **end if**
8: Update policy via: $\mathcal{J}_{\text{SuperRL}}(\theta) = E_{\tau \sim \pi_\theta}[r(\tau \mid q)] - \mu(q)\,\text{KL}(\pi_\beta \| \pi_\theta)$

---

**Caption:** SuperRL selectively triggers SFT under sparse reward or advantage conditions, using supervised signals from reference policies to guide learning whenever reinforcement feedback is insufficient. This ensures stable policy improvement even in low-signal regimes. The policy is updated using: $\mathcal{J}_{\text{SuperRL}}(\theta) = \mathbb{E}_{\tau \sim \pi_\theta}[r(\tau \mid q)] - \mu(q)\,\text{KL}(\pi_\beta \| \pi_\theta)$.

In essence, SFT acts as a safety net in signal-poor regimes, while RL continues to guide policy improvement wherever possible.

**SuperRL Variants.** To clarify the triggering mechanism, we define two variants of SuperRL based on the type of sparse signal that activates SFT:

- **SuperRL-R:** SFT is triggered whenever the reward is zero. This is the default behavior.

- **SuperRL-A:** SFT is triggered whenever the advantage is zero, allowing finer-grained control based on trajectory-specific improvements.

These variants provide flexibility in handling different types of sparse or uninformative RL signals, while maintaining the adaptive safety net property of SFT.

## 4 EXPERIMENTS

### 4.1 EXPERIMENTAL SETUP

To systematically evaluate SuperRL, we design experiments under two complementary settings that target distinct aspects of model capability.

**Base model setup.** For base LLMs, our goal is to isolate the effect of adaptive SFT triggering under controlled conditions. We therefore train on a single reasoning dataset and evaluate across multiple benchmarks. This setup minimizes confounding factors by simplifying the training environment while stressing cross-task generalization. In this regime, we aim to answer whether SuperRL can extract stable performance gains from limited training data and transfer effectively across diverse evaluation tasks.

**Math model setup.** For math-specialized LLMs, our goal is to evaluate the scalability of SuperRL under heterogeneous supervision conditions. We conduct experiments on multi-task training corpora that encompass diverse reasoning datasets with varying difficulty levels, and evaluate performance on held-out mathematical reasoning benchmarks. This setting enables us to comprehensively assess whether SuperRL can effectively handle diverse task structures, adapt across difficulty regimes, and maintain robust generalization in complex reasoning domains.

**Evaluation protocol.** For the base model scenario, we focus on two aspects: (1) whether SuperRL yields consistent improvements across multiple benchmark datasets with different difficulties, and (2) how the SuperRL variants differ in adaptation efficiency and generalization under controlled training conditions.

For the math-specialized scenario, we emphasize scalability and competitiveness: (1) testing whether SuperRL sustains robust performance under heterogeneous supervision and increasing reasoning complexity, and (2) comparing against state-of-the-art integration methods to quantify its effectiveness.

Further details of datasets, models, training settings, and evaluation procedures are provided in the Appendix D.

## 4.2 INSIGHTS FROM SINGLE-DATASET TRAINING OF SUPERRL

Table 1: Cross-dataset generalization accuracy (%) for models trained on a single reasoning dataset using the **Qwen2.5-1.5B** base model. Each block lists results for one training dataset (leftmost column) and compares methods: *RL-only* (reinforcement-learning baseline), *SFT-only*, *SFT-then-RL*, *SuperRL-R* and *SuperRL-A*. The **In Distribution** column reports accuracy on the held-out test set of the *training* dataset (i.e., the in-distribution test). The **Out Of Distribution** columns show accuracy when the same model is evaluated on other benchmarks (GSM8K, PRM12K, OpenR1) and on the competition sets **AIME24** and **AIME25**; entries marked "–" are not applicable because they would duplicate the in-distribution evaluation. Boldface indicates the best accuracy for each specific train–test pair.

| Training | Method | In Distribution | Out Of Distribution | | | | |
|---|---|---|---|---|---|---|---|
| | | Training dataset | GSM8K | PRM12K | OpenR1 | AIME24 | AIME25 |
| GSM8K | RL-only | 72.29 | - | 33.18 | 14.96 | 2.92 | 1.10 |
| | SFT-only | 58.00 | - | 22.86 | 7.14 | 0.00 | 0.00 |
| | SFT-then-RL | 71.45 | - | 32.14 | 14.58 | 1.45 | 0.44 |
| | SuperRL-R | **78.86** | - | 36.96 | **16.11** | **4.10** | 1.81 |
| | SuperRL-A | 76.43 | - | **37.48** | 15.50 | 3.46 | **2.20** |
| PRM12K | RL-only | 44.48 | 64.61 | - | 19.65 | 2.59 | 2.09 |
| | SFT-only | 22.86 | 61.43 | - | 11.43 | 0.00 | 0.00 |
| | SFT-then-RL | 42.99 | 67.56 | - | 16.61 | 2.71 | 1.21 |
| | SuperRL-R | 48.10 | 70.95 | - | 19.35 | **4.28** | 2.51 |
| | SuperRL-A | **51.00** | **74.90** | - | **20.17** | 2.26 | **2.51** |
| OpenR1 | RL-only | 17.40 | 56.98 | **42.10** | - | 2.91 | 0.99 |
| | SFT-only | 11.43 | 65.00 | 28.57 | - | 3.33 | 0.00 |
| | SFT-then-RL | 14.93 | 66.77 | 30.66 | - | 3.04 | 1.42 |
| | SuperRL-R | **19.68** | **73.68** | **42.10** | - | **3.55** | **1.84** |
| | SuperRL-A | 10.23 | 14.28 | 28.57 | - | 0.56 | 0.74 |

### 4.2.1 MAIN RESULTS

We conducted experiments using the Qwen2.5-1.5B model to explore the performance of SuperRL under single-dataset training, where models were trained individually on three reasoning datasets (**GSM8K**, **PRM12K**, and **OpenR1**) and evaluated on both in-distribution (ID) held-out sets and out-of-distribution (OOD) benchmarks (including cross-dataset tasks and high-difficulty mathematical competition datasets **AIME24/AIME25**). The goal was to validate whether SuperRL, through its

adaptive integration of SFT and RL, could outperform traditional training paradigms in both basic task performance and generalization capability.

**Dataset Selection.**    These datasets were carefully chosen to cover a spectrum of task difficulty for the Qwen2.5-1.5B model. From the ID evaluation scores, one can observe a progression from relatively easy tasks (**GSM8K**) to more challenging ones (**PRM12K** and **OpenR1**). This diversity allows us to assess the robustness of SuperRL under both **dense-reward scenarios** (where feedback is frequent and learning signals are abundant) and **sparse-reward scenarios** (where feedback is limited and learning is more challenging), demonstrating that the method provides consistent benefits across varying difficulty levels. In this sense, the datasets serve not only as benchmarks but also as proxies for environments with different reward sparsity, highlighting the general applicability of our approach.

**In-Distribution Performance.**    Across all scenarios, SuperRL consistently led—whether tasks were ID or OOD. Quantitatively, the improvements were striking: In ID evaluations, GSM8K-trained **SuperRL-R** achieved **78.86%** accuracy, 6.57 percentage points higher than RL-only (72.29%). For PRM12K, **SuperRL-A** hit **51.00%** accuracy, more than doubling SFT-only's performance (22.86%) and outperforming RL-only by 6.5 percentage points.

**Out-of-Distribution Performance.**    In OOD tasks, the gains were even more notable. OpenR1-trained **SuperRL-R** reached **73.68%** accuracy on GSM8K (OOD), a dramatic 16.7-percentage-point jump over RL-only (56.98%). Similarly, PRM12K-trained **SuperRL-A** achieved **74.90%** on OOD GSM8K, 10.29 percentage points higher than RL-only (64.61%). Even on high-difficulty **AIME24**, GSM8K-trained **SuperRL-R** delivered measurable gains, outperforming RL-only by 1.18 percentage points in a task where traditional methods struggled. These significant improvements validate that SuperRL's adaptive design effectively addresses the limitations of single-paradigm training, boosting both task-specific performance and generalization.

These significant improvements validate that SuperRL's adaptive design effectively addresses the limitations of single-paradigm training, boosting both task-specific performance and generalization. In addition, we also analyze the reward dynamics obtained on both training and test sets during model training, with detailed results provided in the Appendix G.

### 4.2.2    SuperRL-R vs. SuperRL-A: Dependency on Data Quality

An important factor influencing the performance of SuperRL is the quality of the training dataset. As shown in Table 1, the impact of dataset quality becomes particularly evident when comparing **SuperRL-R** and **SuperRL-A**.

**Dependency on Expert Data.**    SuperRL-A heavily relies on high-quality expert data throughout both early and late stages of training. This reliance can be interpreted in terms of reinforcement learning signals: the early stage primarily uses dense advantage signals derived from SFT (advantage-rich guidance), while the later stage still depends on SFT when the RL reward is sparse (reward-sparse scenarios). In other words, when task feedback is limited or delayed, SuperRL-A continues to lean on expert data to provide informative guidance.

In contrast, SuperRL-R mainly leverages expert data during the early stage, providing a cold-start advantage for policy initialization. After this initial phase, RL signals dominate, allowing the model to adapt without interference from potentially low-quality SFT examples. This distinction explains why SuperRL-A is more sensitive to dataset quality, especially in sparse-reward environments, while SuperRL-R exhibits more stable performance across different data scenarios.

**Effect on Different Datasets.**    The influence of dataset quality aligns with our earlier ranking based on answer correctness, step completeness, and numerical reliability: **GSM8K** (highest quality, fully human-verified, logically rigorous) > **PRM12K** (high quality, model-generated with human validation, occasional minor errors) > **OpenR1** (largest scale, fully model-generated with minimal human verification, occasional logical or numerical errors). This ranking reflects both the dataset construction methodology and the resulting answer reliability.

Table 2: Entropy statistics of RL and SuperRL across selected datasets. Smaller variance indicate more stable output distributions.

| Method & Dataset | Max | Min | Range | Variance |
|---|---|---|---|---|
| RL-only (GSM8K) | 4.850 | 0.004 | 4.846 | 0.171 |
| SuperRL (GSM8K) | 4.875 | 0.006 | 4.869 | 0.134 |
| RL-only (PRM12K) | 5.102 | 0.012 | 5.090 | 0.512 |
| SuperRL (PRM12K) | 5.086 | 0.010 | 5.076 | 0.441 |
| RL-only (OpenR1) | 4.978 | 0.044 | 4.934 | 0.708 |
| SuperRL (OpenR1) | 5.208 | 0.034 | 5.174 | 0.695 |

On simpler datasets, where reasoning patterns are relatively straightforward, SuperRL-A performs roughly on par with SuperRL-R, as both methods can efficiently learn from available signals and adaptive SFT provides limited additional benefit. However, on more challenging datasets, where reasoning complexity is higher, SuperRL-A can fully leverage high-quality offline expert data through adaptive SFT. This enables it to better imitate correct reasoning patterns and potentially outperform SuperRL-R, particularly when expert guidance is dense and reliable.

Conversely, on lower-quality datasets such as OpenR1, the continued reliance of SuperRL-A on noisy SFT data can impede learning, leading to suboptimal performance. SuperRL-R, by contrast, limits SFT usage to the early stage and subsequently relies on RL signals, avoiding interference from lower-quality data and achieving more stable improvements across datasets.

**Takeaways.** Based on both dataset quality and our empirical results, we summarize the main insights as follows:

- **SuperRL-R is the safer default.** It consistently improves performance across datasets of varying quality, providing robust gains even when SFT data is noisy, while still benefiting from high-quality expert guidance during early training.

- **SuperRL-A excels in challenging tasks with high-quality expert data.** When tasks are complex and dense, reliable offline guidance is available, adaptive SFT allows SuperRL-A to better imitate correct reasoning patterns and potentially outperform SuperRL-R.

- **Dataset quality and task difficulty jointly determine the relative advantage.** On simpler datasets or when expert data is noisy, SuperRL-R's early-stage SFT plus RL-dominated learning ensures more stable results, whereas SuperRL-A's reliance on continued SFT can hurt performance.

In addition, we provide a detailed discussion on when to adopt the *SFT-then-RL* training paradigm in Appendix E.

### 4.2.3 STABILITY OF SUPERRL

Beyond accuracy improvements, an important dimension of evaluation is the *stability* of the training process. Instability in reinforcement learning often manifests as oscillating predictions, inconsistent action choices, or highly variable reasoning steps across similar inputs. To probe this aspect, we analyze the entropy statistics of model output distributions. Here, **stability is reflected not in absolute entropy values, but in the magnitude of entropy fluctuations**: smaller *range* and lower *variance* imply that the model makes predictions with more consistent confidence across samples.

**Entropy variation as stability evidence.** Compared to vanilla RL, SuperRL typically exhibits **smaller range and reduced variance**, meaning its predictions fluctuate less in confidence across samples. For example, on HITAB, the variance decreased from 0.363 to 0.186 and the variance dropped by nearly 50%, showing that the model's output distribution is more consistent and less erratic under SuperRL training. This stabilization effect is also visible on GSM8K and PRM12K, where variance reductions suggest that SuperRL helps the model maintain steadier reasoning trajectories.

**Dataset-dependent trends.** The magnitude of entropy fluctuation reveals how dataset quality interacts with stability. On high-quality datasets such as GSM8K and PRM12K, SuperRL substantially lowers entropy variance, indicating more reliable reasoning across inputs. On noisier datasets like OpenR1, the maximum entropy may increase due to noisy SFT guidance, yet both the range and variance shrink compared to RL-only, suggesting that the adaptive mechanism still dampens instability even when supervision is imperfect.

**Takeaway.** These results show that SuperRL improves stability by reducing the *amplitude of entropy fluctuations*, rather than by uniformly lowering entropy values. This means that SuperRL-trained models are less prone to sharp confidence swings across inputs, yielding more reliable and predictable behavior—a critical property for robust deployment in complex reasoning domains.

### 4.3 Insights from Hybrid-Dataset Training of SuperRL

The DeepScaleR(Luo et al., 2025) report attributes its strong performance to three main factors: (1) **Model advantage** — it builds on **DeepSeek-R1-Distill-1.5B**, a distilled reasoning-specialized model, while our earlier experiments used vanilla **Qwen2.5-1.5B**; (2) **Data scale** — DeepScaleR is trained on ∼40k curated pairs (AIME, AMC, Omni-MATH, Still), far larger than datasets such as `LIMO` (817 samples); (3) **Pipeline design** — a two-stage RL process combining large-scale supervised preference optimization with later dynamic reward switching, supported by ∼3,800 A100 GPU hours (∼ \$4,500).

To provide a fairer comparison, we replicated Stage 1 of DeepScaleR due to limited resource using the official dataset and the same base model. Despite operating under tighter compute and memory budgets (e.g., using only half the output context window size), our SuperRL-Stage1 achieves performance nearly identical to the official DeepScaleR Stage 1. The results are summarized in Table 3.

Table 3: Comparison on AIME 2024 (% accuracy). SuperRL-Stage1 (ours) achieves performance comparable to DeepScaleR-Stage1 despite limited compute.

| Model | AIME 2024 | Output Max Length |
|---|---|---|
| Qwen-2.5-Math-7B-Instruct | 13.3 | 8K |
| rStar-Math-7B | 26.7 | 8K |
| Eurus-2-7B-PRIME | 26.7 | 8K |
| Qwen2.5-7B-SimpleRL | 26.7 | 8K |
| DeepSeek-R1-Distill-1.5B | 28.8 | 8K |
| Still-1.5B | 32.5 | 8K |
| DeepScaleR-Stage1 | 33.9 | 8K |
| **SuperRL-Stage1 (Ours)** | **33.3** | **4K** |

As shown in Table 3, SuperRL-Stage1 attains 33.3% accuracy, essentially matching DeepScaleR-Stage1 (33.9%) while surpassing several larger 7B-scale baselines. This demonstrates that SuperRL can deliver near state-of-the-art reasoning performance under highly constrained computational resources, highlighting its efficiency and robustness.

Beyond DeepscaleR, we further conducted a series of experiments on diverse hybrid datasets, which consistently validated the effectiveness of our method when compared against recent approaches such as LUFFY(Yan et al., 2025) and ReLIFT(Ma et al., 2025). These broader results, together with an extended discussion of comparison results, are provided in Appendix I.

## 5 Conclusion

We present SuperRL, a unified training framework that adaptively combines supervised and reinforcement signals to improve reasoning under both dense and sparse rewards. Experiments show that SuperRL achieves superior performance, stability, and generalization across diverse reasoning benchmarks. Limitations and future work are discussed in Appendix B.

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

## A    LLM Usage

In preparing this work, we made limited use of large language models (LLMs) as a writing and editing assistant. Specifically, an LLM was used to improve the clarity and readability of several sections by suggesting alternative phrasings, reorganizing sentences for better flow, and checking for grammatical consistency. All core ideas, experimental design, data analysis, and conclusions were developed entirely by the authors. The authors carefully reviewed and verified all LLM-assisted edits to ensure accuracy and originality.

## B    Limitations and Future Work

While SuperRL achieves strong empirical performance across diverse reasoning benchmarks, several limitations remain.

First, the fallback mechanism is inherently reactive: it triggers supervised fine-tuning (SFT) only when all sampled trajectories yield zero reward. Although this avoids overwriting useful gradients from positive rollouts, it does not proactively exploit offline data to guide early exploration or counteract suboptimal trajectory distributions. More proactive fallback strategies—such as leveraging reward-prediction confidence, entropy-based thresholds, or offline pre-filtering—could improve both stability and sample efficiency.

Second, although SuperRL dynamically alternates between RL and SFT at the instance level, the two optimization objectives are only loosely coupled, with distinct update rules. This decoupling may weaken credit assignment and hinder smooth gradient integration. A promising direction is to design unified objectives or joint loss formulations that softly interpolate between SFT and RL signals, potentially guided by reward magnitude or rollout quality. From an actor–critic perspective, SuperRL currently applies fallback and reward-guided updates primarily on the actor side, leaving the critic relatively underutilized. Incorporating SFT-style signals or offline supervision into the critic—such as calibrating value estimates with expert demonstrations or confidence-weighted rollouts—could improve reward shaping, reduce variance, and stabilize training, ultimately enabling more effective policy updates on the actor.

Third, our approach assumes the availability of high-quality expert trajectories for fallback SFT. In practice, such data may be scarce or domain-limited. Future extensions may consider bootstrapping fallback supervision from model self-refinement, synthetic data generation, or retrieval-augmented reasoning.

Finally, our evaluation focuses on static reasoning tasks with fixed inputs. Extending SuperRL to interactive or agentic environments—where reasoning involves tool use, memory, or long-horizon planning—introduces challenges such as non-stationarity, temporal abstraction, and more complex credit assignment. Integrating SuperRL with test-time search or planning algorithms (e.g., MCTS or beam-guided rollouts) also represents an exciting avenue for advancing adaptive reasoning under sparse rewards.

In summary, while SuperRL offers a practical and effective framework for unifying supervised and reinforcement-based reasoning, it also highlights a broader design space for adaptive training paradigms that integrate exploration, supervision, and search.

## C    Broader Impacts and Safeguards

As language models become increasingly capable of performing complex reasoning tasks, they are poised to influence high-stakes domains such as education, scientific discovery, legal analysis, and decision support. SuperRL, by improving the reasoning robustness of LLMs, could further accelerate this trend. However, such capabilities also introduce new societal risks, especially if models generate persuasive yet flawed reasoning or learn undesirable behaviors from biased feedback.

A key benefit of SuperRL lies in its fallback to high-quality supervised trajectories when reward signals are absent. This acts as a built-in safeguard, anchoring learning to human-verified reasoning paths and reducing the likelihood of reinforcement-induced divergence. However, reliance on offline data also inherits its limitations—such as annotation bias, narrow coverage, or stylistic homogeniza-

tion—which could be amplified by repeated reuse. Careful curation, diversity analysis, and auditing of demonstration datasets are therefore crucial to mitigate these effects.

Moreover, since SuperRL adaptively integrates reinforcement learning, it inherits known risks from RL-based training: reward hacking, undesired generalization, and non-transparent credit assignment. Although our instance-level switching reduces over-optimization on spurious reward signals, stronger safeguards—such as adversarial evaluation, uncertainty estimation, or reward model interpretability—are needed to ensure alignment in more open-ended settings.

From a deployment perspective, reasoning models trained with SuperRL should be clearly scoped and audited before use in sensitive applications. Developers should monitor not only final accuracy but also behavioral changes introduced by the RL component, especially in failure cases. Human-in-the-loop evaluation, scenario red-teaming, and counterfactual probing can provide additional oversight.

Lastly, while SuperRL improves reasoning competence, it does not address deeper epistemic limitations of LLMs, such as lack of verifiability or causal grounding. Future safeguards may benefit from integrating structured verification, external tool use, or modular reasoning pipelines to augment the transparency and controllability of these systems.

## D SETUP DETAILS OF EXPERIMENTS

### D.1 DATASET CONFIGURATIONS

To rigorously evaluate SuperRL, we consider datasets from two distinct usage scenarios: (1) **Training-only datasets**, which are used for supervised training or preference learning in isolation; (2) **Evaluation-only datasets**, which are reserved purely for benchmarking model performance under zero-shot or few-shot settings.

**Training Datasets.** **GSM8K** Cobbe et al. (2021) is a benchmark of 8.5k human-written grade school math problems, widely used for training models on multi-step arithmetic reasoning. **PRM12K** Lightman et al. (2023) contains 12k math problems, each with five diverse solution paths (both correct and incorrect), making it suitable for preference learning and reward modeling. **OpenR1-Math-220k** is a large-scale dataset of 220k math problems, each with multiple reasoning traces verified for correctness, designed for training models to follow and replicate reasoning processes. **HiTab** Cheng et al. (2021) is a dataset of hierarchical tables and QA pairs, used for training models on numerical and structural reasoning beyond pure math problems. **DeepScaleR-Preview** is a preview dataset from the DeepScaleR project ( 40.3k math problems with problem, solution, answer), under MIT license; only a training split is available.

**Evaluation Datasets.** **AIME 2024** (Li et al., 2024) and **AIME 2025** (Li et al., 2024) are competition-level math benchmarks, each consisting of 30 Olympiad-style problems. They are strictly used as test sets to evaluate symbolic and geometric reasoning. AIME 2025 further introduces diagram-based problems (e.g., TikZ-rendered figures), making it more challenging for spatial reasoning.

Table 4: Datasets used in this study, grouped by usage scenario. "–" indicates the split is not officially provided.

| Dataset | # Train | # Test | Task Type | Domain | License | Source |
|---|---|---|---|---|---|---|
| **Training Datasets** | | | | | | |
| GSM8K | 7,473 | 1,319 | Math word problems | Elementary math | MIT | Link |
| PRM12K | 12,000 | – | Preference reasoning | Mathematics | Apache 2.0 | Link |
| OPENR1-MATH-220K | 220,000 | – | Math reasoning | Mathematics | Apache 2.0 | Link |
| HITAB | 7,399 | 1,583 | Hierarchical table QA | Statistics | C-UDA 1.0 | Link |
| DEEPSCALER-PREVIEW | 40,300 | – | Math reasoning | Mathematics | MIT | Link |
| **Evaluation Datasets** | | | | | | |
| AIME 2024 | – | 30 | Competition math | Olympiad | MIT | Link |
| AIME 2025 | – | 30 | Competition math | Olympiad | CC BY 4.0 | Link |

## D.2 Model Configurations

We evaluate SuperRL on a diverse set of open-source language models spanning multiple families and parameter scales. Specifically, we consider three representative families: **Qwen2.5** (Yang et al., 2024a), **DeepSeek-R1 Distilled** (Guo et al., 2025), and **LLaMA 3.x**. Together, they cover models from 0.5B to 8B parameters, enabling systematic analysis of both scale effects and architectural differences.

**Qwen2.5** models (0.5B, 1.5B, 3B, 7B) provide a consistent backbone for studying reasoning performance under scaling. **DeepSeek-R1-Distill-1.5B** serves as a compact reasoning-focused model distilled from a larger teacher. **LLaMA 3.x** models (1B, 3B, 8B) represent widely adopted community baselines with strong instruction-following ability.

All models are publicly available under permissive licenses, ensuring reproducibility. Table 5 summarizes the configurations.

Table 5: Models evaluated in this study.

| Model | Parameters (B) | Family |
|---|---|---|
| Qwen2.5-0.5B | 0.5 | Qwen2.5 |
| Qwen2.5-1.5B | 1.5 | Qwen2.5 |
| Qwen2.5-3B | 3 | Qwen2.5 |
| Qwen2.5-7B | 7 | Qwen2.5 |
| DeepSeek-R1-Distill | 1.5 | DeepSeek-R1 |
| LLaMA 3.1-1B | 1 | LLaMA 3.x |
| LLaMA 3.1-3B | 3 | LLaMA 3.x |
| LLaMA 3.1-8B | 8 | LLaMA 3.x |

## D.3 Training Environment Setup

All experiments are conducted within the `verl` framework, a scalable actor–critic reinforcement learning platform tailored for optimizing language models. This framework serves as the foundation for our experimental setup, allowing us to implement and iterate on various training strategies. Our primary experiments utilize GRPO as the core reinforcement learning algorithm. However, to validate the general applicability of our uncertainty-weighted hybrid training framework, we also conduct comparative trials using PPO. The configurations, scripts and source code for these experiments are all developed and modified within the `verl`. This includes the implementation of the GRPO and PPO algorithms, as well as the integration of the uncertainty-weighted hybrid training approach. The flexibility of the `verl` enables us to seamlessly update and refine our methods, ensuring that our experiments are both robust and adaptable.

Our experimental scripts follow a unified and modular configuration framework designed to ensure consistency across all datasets and model backbones. This framework supports flexible adaptation, with minor adjustments made to accommodate variations in model scale (e.g., parameter count) and available computational resources (e.g., GPU memory capacity). By default, both the actor and critic are initialized from the same pretrained checkpoint. This shared initialization helps maintain stability in the early stages of training and ensures that policy updates build upon a consistent starting point. Unless otherwise noted, we train the entire model with full-parameter updates, rather than using techniques like partial tuning or adapters. To manage memory consumption during training, we enable gradient checkpointing, which trades off additional computation for significantly reduced memory usage. This is particularly important when training large models with long sequences or large batch sizes. To constrain policy drift and encourage stable learning, we apply KL divergence regularization between the current policy and a fixed reference policy. We use a low-variance KL formulation with a fixed regularization coefficient of 0.001, following prior work showing its effectiveness in language model fine-tuning.

We adopt a fixed learning rate of 1e-6 for all experiments, regardless of dataset or model size. This consistent setting simplifies hyperparameter tuning and facilitates fair comparisons across different tasks. The batch size is set to 32 by default, which provides a good balance between training stability

and GPU memory efficiency. However, in data-scarce scenarios—such as the LIMO dataset, which contains relatively few high-quality training samples—we reduce the batch size to 8 to improve gradient quality and mitigate overfitting risks associated with small datasets. Each training run proceeds for 500 update steps, a schedule empirically chosen to ensure sufficient optimization while maintaining computational efficiency. To monitor progress and detect training instabilities early, we conduct model evaluation every 5 steps on a held-out validation set. At the end of training, we report the test-set performance at step 500 if the learning curve shows smooth convergence. In cases where the validation curve exhibits fluctuations or noise—typically due to reward sparsity or instability in policy gradients—we apply exponential moving average (EMA) smoothing to the score trajectory to obtain a more reliable final evaluation metric. This ensures that our reported performance reflects the overall trend rather than being biased by momentary spikes or drops.

For rollout generation, we employ the `vLLM` backend, which enables efficient and scalable batched decoding with optimized GPU memory usage. Each prompt is decoded to produce five candidate responses, allowing for diverse sampling during training. To ensure stable runtime behavior, we cap GPU memory utilization at 40%. The decoding configuration adopts a stochastic sampling strategy with a temperature of 1.0, top-$k = -1$ (disabled), and top-$p = 1.0$, corresponding to unconstrained nucleus sampling. These settings encourage diverse yet coherent response generation from the model. We constrain the maximum sequence length—comprising both the input prompt and the generated output—to 2048 tokens. Prompts that exceed this limit are filtered out prior to generation, and any attempt to exceed the limit during decoding raises a truncation error. This strict enforcement helps maintain consistency and prevents the introduction of ambiguous or malformed training signals.

For dataset partitioning, we follow the original train/test splits provided by the benchmark for GSM8K and HiTab, ensuring compatibility with prior work and fair comparison. For all other datasets containing more than 20,000 examples, we randomly subsample a total of 20,000 instances to reduce computational cost while maintaining representative coverage. The selected subset is then split into training and test sets using an 80/20 ratio. This standardized partitioning protocol facilitates consistent evaluation across diverse datasets with varying sizes and distributions.

All experiments are conducted on machines equipped with NVIDIA H100 GPUs. For most settings involving smaller models (e.g., 1-3B parameters), we utilize a 2-GPU configuration, which provides sufficient compute capacity for full-parameter training. In contrast, larger models (e.g., 7B and above) are trained in a distributed fashion using multiple nodes and GPUs. We construct multi-node clusters using `Ray`, a flexible framework for large-scale distributed computing. Once initialized, training is orchestrated through `verl`'s distributed utilities, which support scalable actor–critic reinforcement learning with efficient inter-GPU communication and synchronization. The micro-batch size is fixed at 2 per GPU across all experiments, regardless of the number of nodes or model size. This setting balances memory usage and gradient estimation stability, especially under reinforcement learning with sparse rewards. To ensure a fair comparison across different training paradigms, we adopt a unified optimizer configuration—including learning rate, weight decay, and scheduler—for all methods: supervised-only (SFT), reinforcement learning-only (RL), sequential SFT+RL, and our proposed hybrid RL+SFT. This design isolates the effect of training strategies from confounding optimization differences.

For the SFT and SFT+RL baselines, we begin by fine-tuning the base model using the same training dataset, learning rate, and context length as in the RL-based training setups. The supervised fine-tuning is conducted for a total of 25 epochs, a duration chosen to ensure sufficient convergence without overfitting. Throughout training, we evaluate each epoch's checkpoint on the held-out test set using greedy decoding, i.e., with temperature set to zero and no sampling. The checkpoint achieving the highest test-set performance is selected as the final SFT baseline. For the SFT+RL baseline, reinforcement learning is initialized from this best-performing SFT checkpoint. We then continue training the model under the same RL framework used in our hybrid method. The final performance of the SFT+RL model is reported based on the test-set score at the point where the learning curve reaches stable convergence. If the learning dynamics are noisy, we apply exponential moving average (EMA) smoothing to determine the final score in a robust manner.

### D.4 PROMPT AND OUTPUT FORMAT DESIGN

To promote interpretable and verifiable reasoning behavior, we adopt distinct prompting and output formatting strategies tailored to the model type.

**Prompt Design.** For both vanilla and structured-output models, we append a concise instruction—"Let's think step by step and output the final answer in `boxed{}`."—to the original problem description to encourage step-by-step reasoning and a clearly identifiable final answer.

For example:

> **Prompt:** If you have 3 apples and you buy 2 more, how many do you have? Let's think step by step and output the final answer in boxed{}.

For **vanilla models** without structured output conventions, the model is directly given the above prompt string. A typical output is:

```
We start with 3 apples.  Buying 2 more gives us 3 + 2
= 5.  The final answer is boxed{5}.
```

In contrast, for **structured-output models** (e.g., Qwen2.5-1.5B, DeepSeek-R1-Distill-Qwen-1.5B) that support chat-style prompting, we apply `apply_chat_template` to transform the prompt into a ChatML-formatted conversation. For instance:

```
<|im_start|>system
You are a helpful assistant.<|im_end|>
<|im_start|>user
If you have 3 apples and you buy 2 more, how many do
you have?  Let's think step by step and output the
final answer in boxed{}.<|im_end|>
<|im_start|>assistant
<think>
```

**Expected Model Output and Postprocessing.** The output format likewise depends on the model's interface and training. For structured-output models, the expected output includes special tags:

```
We start with 3 apples.  Buying 2 more gives us 3 + 2
= 5.  Now let's output the final answer.</think>
<answer>The answer is boxed{5}</answer>
```

In this case, we apply tag-aware parsing during postprocessing:

- Extract the reasoning trace enclosed in `<think>` tags.
- Extract the final answer from within the `<answer>` tag, specifically the content of `boxed{}`.

For vanilla models, which do not include tags, we rely on regex-based postprocessing:

- Identify the reasoning portion as any content before the appearance of `boxed{}`.
- Parse the numerical answer from within `boxed{}`.

These model-aware prompt formatting and output parsing strategies ensure the consistent interpretation of model responses across evaluation and training pipelines. They also enable structured reward computation and answer matching for reinforcement learning optimization.

### D.5 REWARD DESIGN

Given a model response $y$ to input $x$, we compute the reward as a binary signal:

$$r(x, y) = \begin{cases} 1 & \text{if extract}(y) = y^{\text{gt}} \\ 0 & \text{otherwise} \end{cases}$$

where extract($y$) denotes the parsed answer obtained via model-specific extraction logic, and $y^{\text{gt}}$ is the ground-truth answer.

To accommodate surface-form variations—especially prevalent in datasets such as LIMO and PRM12K—we incorporate a *canonicalization layer* in the reward computation. This module standardizes answer representations by:

- Normalizing numeric formats (e.g., converting fractions to decimals);
- Performing symbolic equivalence checks (e.g., $2x + 4$ vs. $4 + 2x$);
- Unifying variable names, units, or other context-specific notations.

If the canonicalized prediction matches any canonicalized gold reference, we assign a reward of 1.

As a fallback mechanism, when no delimiters (e.g., \boxed{}) are detected in the model output, we extract the last numerical span as a proxy for the final answer. All reward functions are implemented as deterministic, stateless modules to ensure reproducibility and compatibility with batched rollout evaluations in PPO and GRPO training pipelines.

## D.6  METRIC DESIGN

We adopt **Exact Match (EM)** accuracy as the primary evaluation metric to assess model performance on reasoning tasks. A prediction is considered correct if the extracted answer exactly matches the ground-truth answer after normalization. This includes removing extraneous formatting, standardizing numerical representations, and optionally applying symbolic simplification when applicable. EM offers a strict yet interpretable signal of end-to-end correctness, effectively capturing whether the model arrives at the correct final solution.

Compared to token-level metrics such as BLEU or ROUGE—which quantify n-gram overlap—EM is more aligned with the discrete nature of most reasoning tasks. Token-based metrics often tolerate superficial similarity while overlooking semantically crucial deviations (e.g., predicting 7.0 instead of 7.5), thus failing to penalize incorrect answers that appear linguistically similar. In contrast, EM enforces a high bar for correctness by requiring exact alignment with the reference answer.

To accommodate datasets with multiple valid reasoning paths or equivalent solutions—such as PRM12K and OpenR1—we extend EM to a *relaxed matching* scheme. Specifically, a prediction is marked as correct if it matches *any* of the acceptable reference answers after canonicalization. This allows for flexibility in surface forms (e.g., equivalent algebraic expressions or unit conversions) while preserving the core requirement of semantic equivalence.

In all cases, the normalization and canonicalization procedures used during EM evaluation are kept deterministic and model-agnostic to ensure reproducibility and fairness. This design choice ensures that the metric remains robust across diverse model architectures and output formats.

## E  DISCUSSION OF HITAB DATASET

Table 6: Cross-dataset generalization accuracy (%) for models trained on Hitab (In Distribution vs. Out of Distribution). **In Distribution (ID)**: Test set is identical to the training dataset; **Out of Distribution (OOD)**: Test set is different from the training dataset. Bold numbers indicate the best-performing method within this training block.

| Training | Method | In Distribution (ID) | Out of Distribution (OOD) | | | | |
|---|---|---|---|---|---|---|---|
| | | Corresponding to training dataset | GSM8K | PRM12K | OpenR1 | AIME24 | AIME25 |
| | RL-only | 30.03 (Hitab) | 72.05 | 26.63 | 15.26 | 2.99 | 0.57 |
| | SFT-only | 33.33 (Hitab) | 64.29 | 33.57 | 11.43 | 0.00 | 0.00 |
| Hitab | SFT-then-RL | **38.64 (Hitab)** | 74.56 | **35.91** | **16.65** | 4.41 | **4.11** |
| | SuperRL-R | 18.47 (Hitab) | 77.40 | 29.29 | 12.45 | 3.33 | 2.08 |
| | SuperRL-A | 34.85 (Hitab) | **78.09** | 29.29 | 13.51 | **5.54** | 0.64 |

As shown in Table 6, models trained with the two-stage paradigm (**SFT-then-RL**) consistently outperform not only pure SFT or pure RL baselines but also the more advanced **SuperRL** variants.

This can be explained from the perspective of data distribution. The Hitab dataset has a **narrow and highly structured distribution**, where most problems require reasoning grounded in tables and discrete operations. Training with SFT alone helps the model capture these regularities but limits generalization beyond Hitab due to overfitting to local patterns. Pure RL, on the other hand, encourages exploration but lacks the inductive bias provided by SFT, often leading to reward-driven shortcuts that fail to preserve the structured semantics.

By contrast, **SFT-then-RL** first anchors the model in Hitab's distribution, then uses RL to refine and expand beyond it, yielding a better trade-off between ID stability and OOD robustness. Compared with **SuperRL**, which assumes a more diverse training distribution (e.g., GSM8K, PRM12K), Hitab's narrow distribution makes aggressive reward alternation less effective and more prone to distributional drift.

**Insight.** These findings suggest that when the training distribution is relatively narrow and domain-specific (e.g., tabular reasoning), a sequential **SFT-then-RL** strategy is more suitable than unified or alternating RL frameworks. In other words, the effectiveness of post-training paradigms is highly **distribution-dependent**: datasets with broad, diverse reasoning patterns benefit more from adaptive SuperRL-style exploration, whereas datasets with narrow distributions are better served by the stability of SFT followed by targeted RL refinement.

# F    ANALYSIS OF SFT TRIGGER PATTERN

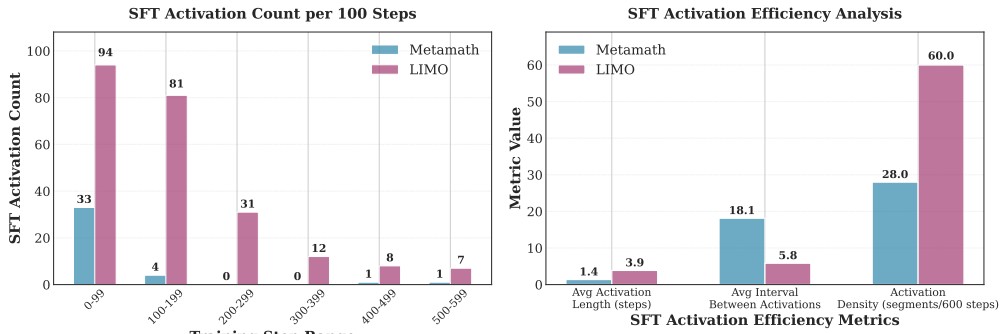

Figure 2: **Analysis of SFT Activation Patterns and Efficiency. Left:** The histogram shows the count of supervised fine-tuning (SFT) activations per 100 training steps for two datasets: *Metamath* and *LIMO*. Most SFT activations in both cases occur in the early phase of training (0–100 steps), with *LIMO* exhibiting a denser activation pattern overall. **Right:** A comparison of SFT activation efficiency metrics across datasets. *LIMO* shows longer average activation lengths, shorter intervals between activations, and a higher activation density (segments with SFT triggered per 500 steps), indicating more frequent and sustained reliance on SFT throughout training.

**SFT Trigger Patterns.**    We observe notable differences in how frequently and efficiently supervised fine-tuning (SFT) is triggered across datasets. As shown in Figure 2, *LIMO* invokes SFT much more often than *Metamath*, especially in the early training stages (0–100 steps), suggesting that it encounters a higher proportion of hard instances requiring fallback supervision. Moreover, *LIMO* exhibits longer and denser SFT activation periods with shorter intervals between activations, indicating greater reliance on offline signals to sustain progress. By contrast, *Metamath* displays sparse SFT usage, consistent with more stable online exploration.

Importantly, across both datasets, reliance on SFT decreases as training progresses. This pattern highlights the adaptive nature of our method: SFT is heavily used in the early stages to guide learning when the policy lacks competency but gradually fades out as the model improves. This reflects a transition from supervision-driven learning to reward-driven generalization.

Table 7 summarizes average scores and SFT usage. We find a strong negative correlation between SuperRL average score and total SFT triggers ($r = -0.86$, $p = 0.028 < 0.05$), suggesting that fallback is most critical for harder tasks where standard RL struggles. Indeed, in datasets with high

Table 7: Comparison of average performance and the number of SFT-triggered updates across datasets. A higher frequency of SFT generally corresponds to more challenging tasks and larger performance gains from *SuperRL*.

| Dataset | RL Avg. Score | SuperRL Avg. Score | SFT Triggers |
|---|---|---|---|
| OpenR1 | 0.526 | 0.619 | 728 |
| PRM12K | 0.268 | 0.590 | 484 |
| HiTab | 0.645 | 0.665 | 270 |
| LIMO | 0.500 | 0.688 | 247 |
| Metamath | 0.715 | 0.700 | 40 |
| GSM8K | 0.687 | 0.758 | 19 |

SFT fallback (e.g., PRM12K, OPENR1), SuperRL achieves much larger gains over RL. Conversely, in easier datasets (e.g., GSM8K, METAMATH), where SFT is rarely needed, the margin is small or reversed.

Interestingly, RL scores alone show a weaker and statistically insignificant correlation with SFT triggers ($r = -0.64$, $p \approx 0.171$), suggesting that raw RL performance is not a sufficient predictor of fallback reliance. Instead, SuperRL dynamically calibrates SFT usage based on rollout outcomes, invoking supervision selectively even when RL achieves moderately good scores.

Overall, these findings underscore the adaptive value of fallback SFT: it provides essential guidance when RL alone plateaus, enhances performance across hard tasks, and reflects the model's internal learning difficulty, thereby bridging the gap between exploration and convergence.

## G TRAINING GRAPH ANALYSIS

The training dynamics depicted in 3 and 4 offer detailed insights into how SuperRL and RL perform across three datasets under distinct reward frameworks, with all metrics tracked as EMA-smoothed Reward Mean@1 over training steps. In GSM8KTrainingGraph.pdf, which focuses on dense reward scenarios, SuperRL demonstrates a clear and consistent advantage over RL across all datasets. On OpenAI/GSM8K, SuperRL reaches a final Reward Mean@1 of 0.774, marking a 7.9% improvement compared to RL's 0.717. The curves show a steady divergence as training progresses, indicating that SuperRL effectively harnesses the dense reward signals to refine its performance over time. For Meta-Math/MetaMathQA, the gap is more pronounced: SuperRL achieves 0.761, a 13.3% gain over RL's 0.671, suggesting its strong adaptability to mathematical reasoning tasks where dense feedback provides critical step-by-step guidance. Most notably, on GAIR/LIMO, SuperRL delivers a remarkable 229.0% improvement, with a final value of 0.526 versus RL's 0.160, underscoring its ability to thrive in complex reasoning environments with dense reward structures.

Turning to 4, which examines sparse reward settings, SuperRL's performance shows nuanced strengths and a single exception. On OpenAI/GSM8K, it achieves a striking 97.0% improvement, with a final Reward Mean@1 of 0.742 compared to RL's 0.377, highlighting its proficiency in extracting meaningful signals from limited feedback in this dataset. For Meta-Math/MetaMathQA, SuperRL maintains a solid edge, reaching 0.776 (34.2% higher than RL's 0.579), with sustained separation in the curves indicating robust learning under sparse supervision. However, on GAIR/LIMO, SuperRL slightly underperforms, with a final value of 0.323 (-1.8% compared to RL's 0.329), suggesting that the dataset's unique characteristics, when combined with sparse rewards, create a scenario where RL's optimization approach aligns more effectively with the limited feedback.

Together, these graphs from 3 and 4 illustrate that SuperRL generally excels in both dense and sparse reward environments, with its performance gains being particularly significant in dense settings and most sparse scenarios, while also revealing a specific case where RL holds a marginal advantage.

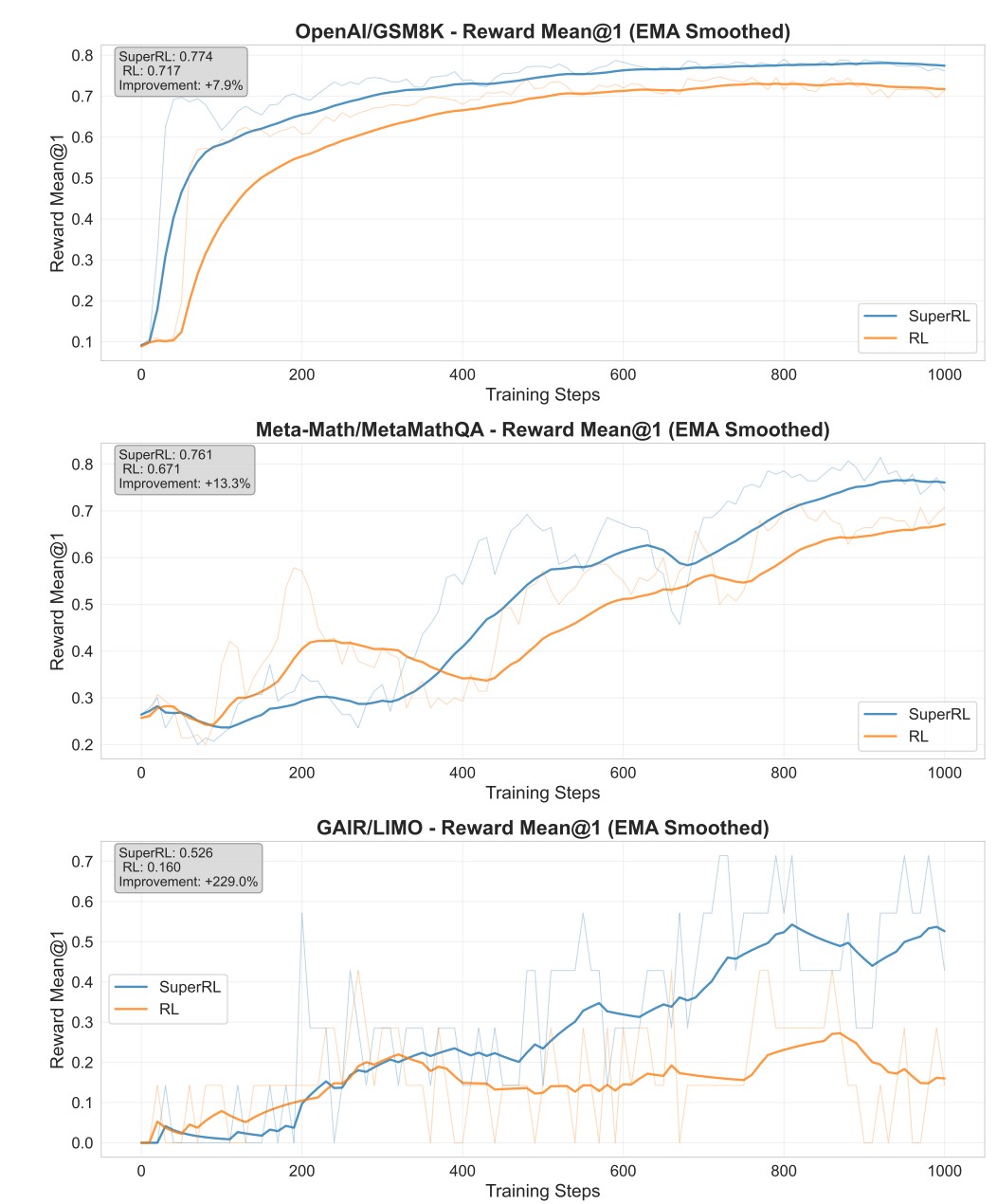

Figure 3: Training curve on the GSM8K dataset with **dense reward**. This graph shows how the performance evolves over time under sparse reward environment.

## H SOFT FUSION OVER HARD SWITCHING: A UNIFIED PERSPECTIVE ON SUPERRL VERSUS SFT+RL

A core challenge in aligning large language models (LLMs) with complex reasoning behaviors lies in how to effectively combine supervised fine-tuning (SFT) and reinforcement learning (RL). While SFT offers high-quality, human-aligned demonstrations, RL allows the model to adapt toward task-specific rewards and discover novel, high-utility behaviors. Yet integrating these two signals remains nontrivial: naïve combinations often result in instability, forgetting, or inefficiency.

The traditional SFT+RL framework adopts a *hard switching* strategy. It first optimizes the model using offline expert data via SFT, and then transitions to on-policy RL to maximize downstream reward

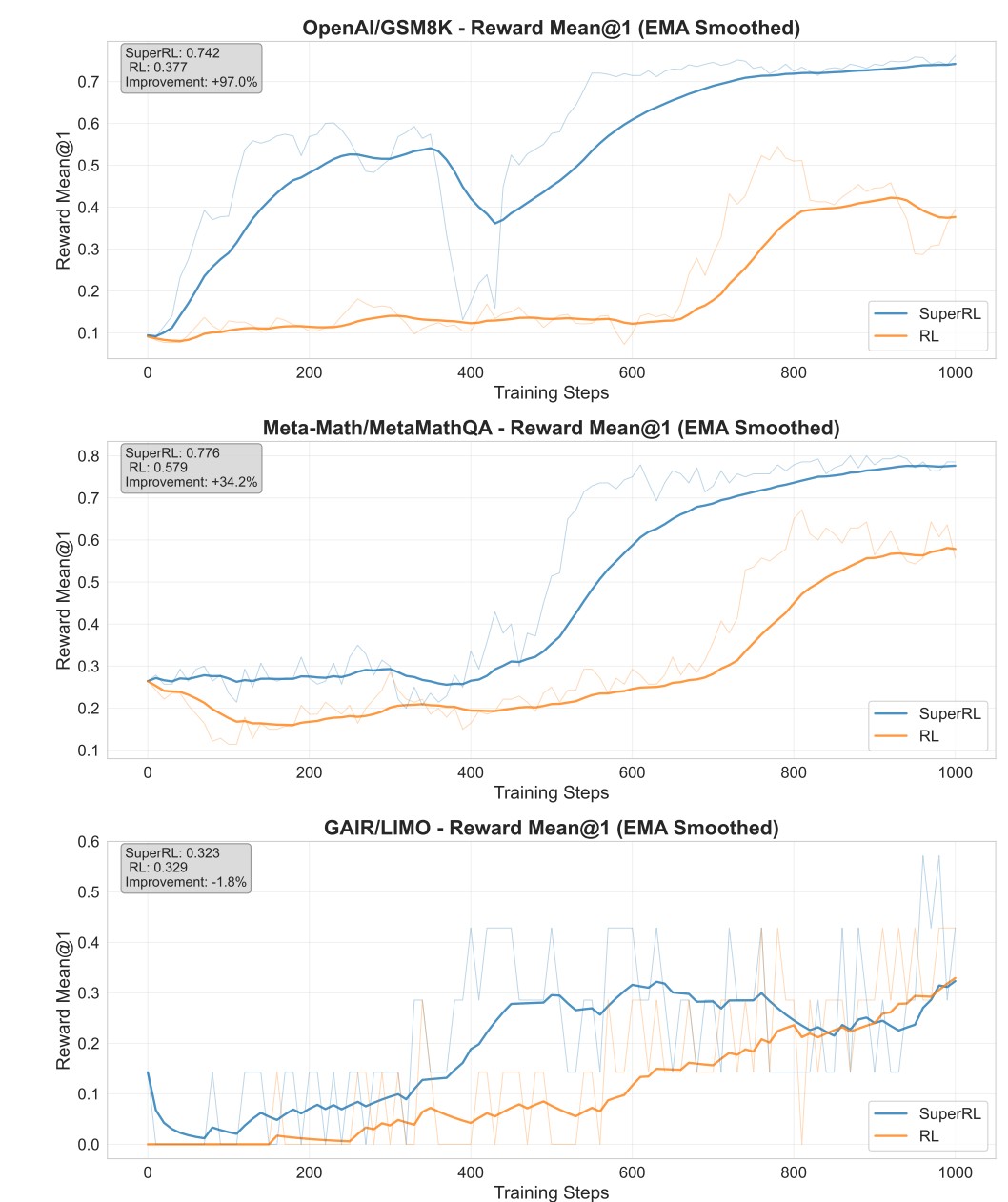

Figure 4: Training curve on the OpenR1 dataset with **sparse reward**. This graph illustrates the challenges and learning dynamics under sparse supervision.

signals. While simple and modular, this two-stage approach suffers from critical limitations. First, the abrupt shift in objectives often causes *catastrophic forgetting* of previously learned behavior, especially under sparse or noisy rewards. Second, once the model is anchored to demonstration-like behaviors, it may resist exploring off-distribution actions that yield higher long-term reward. Third, RL without continual alignment can overfit to narrow reward functions, producing undesired or brittle behavior.

In contrast, the latest version of **SuperRL** implements a *lightweight but principled soft fusion* strategy: for each training instance, the model performs multiple rollouts and examines their rewards. If any rollout receives a non-zero reward, RL updates are applied; otherwise, the model falls back to supervised updates using expert demonstrations. This dynamic instance-level switching seamlessly

Table 8: Comparison of different training strategies on Qwen2.5-Math-1.5B.

| Model | AIME24 | AIME25 | MATH-500 |
|---|---|---|---|
| Qwen-Math-1.5B | 3.0 | 1.4 | 44.4 |
| Qwen-Math-1.5B-Instruct | 8.1 | 6.6 | 66.0 |
| SFT-only | 12.7 | 13.0 | 71.8 |
| RL-only (GRPO) | 10.2 | 8.7 | 71.4 |
| ReLIFT | 13.1 | 8.8 | 73.6 |
| LUFFY | 16.0 | 13.1 | 80.2 |
| SuperRL (ours) | 18.65 | 19.27 | 76.43 |

blends exploration with imitation: rather than predefining a training schedule or manually tuning loss weights, SuperRL adaptively selects the most informative signal—RL or SFT—for each input based on reward feedback.

This formulation brings several practical and conceptual advantages:

- It avoids sharp transitions between learning paradigms and retains the *stability of SFT* throughout training.
- It ensures that online exploration is grounded by offline knowledge in low-reward regions, improving learning in sparse-reward or hard-exploration regimes.
- It introduces no additional architectural complexity—only a reward-gated control flow—making it both simple to implement and scalable across tasks and model sizes.
- It encourages generalization beyond demonstrations by using SFT only when needed, preventing the model from overfitting to static traces.

Compared to previous "soft fusion" approaches that require manually tuned loss interpolation or gradient-level blending, SuperRL leverages reward feedback as a *data-driven switch*, reducing the risk of signal interference while preserving training efficiency. This design effectively bridges the strengths of both paradigms: *SFT provides a fallback anchor when RL fails, and RL enables adaptive generalization when reward signals are informative*.

Nonetheless, SuperRL's strategy is not without limitations. First, it relies on *reward observability*: when rewards are extremely noisy or delayed, the switch decision may become unstable. Second, unlike continuous fusion, the fallback mechanism may not fully capture finer-grained trade-offs between imitation and exploration within the same instance. Third, in domains where reward functions are dense and fully aligned with demonstrations (e.g., synthetic games), pure RL may suffice and converge faster without SFT fallback.

Even so, for open-ended reasoning, long-horizon inference, and weak supervision settings—where reward signals are partial and data distributions shift—SuperRL offers a *robust, generalizable, and efficient alternative to hard-switch SFT+RL pipelines*. It avoids the brittleness of static methods while maintaining a clean training loop and strong empirical performance.

In summary, SuperRL reimagines soft fusion not as continuous loss interpolation but as *instance-level reward-aware fallback*, enabling practical and scalable integration of offline supervision with online exploration. This simple yet effective strategy offers a compelling alternative to traditional hard-switching pipelines, particularly for real-world LLM training under sparse or uncertain feedback.

## I  MORE INTEGRATION RESULTS

Table 8 compares **SuperRL** with prior integration strategies on Qwen2.5-Math-1.5B. The results for LUFFY (Yan et al., 2025) and ReLIFT (Ma et al., 2025), including their reported SFT-only and RL-only baselines, are directly cited from the original papers. We executed experiments for SuperRL on a reduced subset of the same datasets, restricted to the Qwen2.5-Math-1.5B family. The SuperRL row corresponds to our own reproduction, while all other rows are taken from prior reports and papers.

Several observations can be drawn:

- **Baseline methods (SFT-only vs. RL-only).** Pure SFT achieves reasonable performance but fails to adapt beyond demonstrations, while RL-only training shows limited gains and even degradation on AIME25. This highlights the fundamental difficulty of relying on a single signal—SFT offers stability but limited exploration, whereas RL alone struggles with sparse or noisy feedback.

- **Prior integration methods (ReLIFT and LUFFY).** Both ReLIFT and LUFFY attempt to combine the strengths of SFT and RL, but in different ways. ReLIFT leverages iterative preference learning and fine-tuning, yielding modest improvements over the baselines. LUFFY introduces a more sophisticated optimization schedule and achieves stronger results on MATH-500, showing the benefit of carefully designed hybrid training. However, both approaches involve additional training stages and non-trivial implementation complexity, which may limit reproducibility and scalability.

- **SuperRL (ours).** In contrast, SuperRL adopts a lightweight, reward-aware fallback mechanism. Even under resource-constrained reproduction, SuperRL achieves the strongest overall performance: it outperforms all methods on both AIME24 (18.65 vs. 16.0 for LUFFY) and AIME25 (19.27 vs. 13.1 for LUFFY), while remaining competitive on MATH-500 (76.43 vs. 80.2 for LUFFY). Importantly, this is achieved without elaborate multi-stage pipelines or manual loss interpolation—only a simple instance-level control flow that adaptively chooses between RL and SFT updates.

Overall, these results suggest that **SuperRL offers a more practical and generalizable alternative** to existing integration strategies. While LUFFY and ReLIFT demonstrate the potential of hybrid methods, they rely on heavy training schedules or complex design choices. SuperRL, by contrast, achieves superior performance on challenging AIME tasks and competitive results on large-scale benchmarks with a clean, scalable training loop that requires minimal tuning.

**Generalization advantage.** Although SuperRL is slightly behind LUFFY on MATH-500, its substantial improvements on AIME24/25 highlight stronger generalization to small-sample reasoning tasks, where robustness to sparse feedback is especially critical.

