# OpenReview forum: "SuperRL: Reinforcement Learning with Supervision to Boost Language Model Reasoning"
_ICLR.cc/2026/Conference — Submitted to ICLR 2026_

### Official Review · Reviewer_k1zf · 2025-10-25

**Soundness:** 3
**Presentation:** 3
**Contribution:** 2
**Rating:** 4
**Confidence:** 4

**Summary:**

This paper proposes SuperRL, a framework blending RL and SFT adaptively. It uses RL for non-zero rewards, SFT for zero rewards, outperforming baselines in sample efficiency, generalization, and stability across reasoning benchmarks

**Strengths:**

1. The method is simple yet practical, requiring minimal architectural change.

2. The paper is clearly written and well-organized, providing an intuitive explanation of both the theoretical motivation and algorithmic design of SuperRL.

**Weaknesses:**

1. The evaluation scope is somewhat narrow, focusing mainly on mathematical and reasoning benchmarks without validating applicability to broader or interactive tasks such as tool use or dialogue-based reasoning.

2. The novelty of the paper is limited. Although the adaptive fallback is smart, the paper should explain more clearly how it’s different from existing hybrid SFT-RL methods.

3. The theoretical justification should be explained more clearly, with stronger intuition, and clearer connections to the practical SuperRL framework.

**Questions:**

no

---

> ### Author Response · Authors · 2025-11-21
>
> ## Response to Reviewer Comments
>
> We sincerely thank the reviewer for the thoughtful and constructive feedback. Your comments have helped us clarify the design motivations, better articulate theoretical insights, and strengthen the presentation of our work. Below we provide detailed responses to all points raised.
>
> ## Weaknesses
> ### [W1] Narrow Evaluation Scope
>
> We appreciate the reviewer’s concern regarding the breadth of our evaluation. Our experiments primarily target reasoning-centric datasets because they probe logical, mathematical, and general reasoning abilities most directly—areas where SuperRL is designed to improve learning stability and reward usage.
>
> Importantly, our in-house HiTab dataset is fundamentally distinct from typical math datasets: it requires multi-hop reasoning over structured tables involving row/column selection, numerical aggregation, and contextual inference rather than symbolic equation solving. Despite this significant domain gap, SuperRL transfers positively to math benchmarks such as GSM8K, MetaMath, and PRM12K, indicating that the method improves general reasoning ability rather than specializing to a single task type.
>
> We acknowledge that wider evaluations (e.g., tool-use, multi-step dialogue benchmarks) would further strengthen our generalization claims. Due to compute and time constraints, we were unable to run those larger-scale evaluations for the current submission. We are actively securing additional resources and plan to include these broader evaluations in follow-up work. Notably, prior GRPO-based studies also focus on math/reasoning benchmarks, so our current evaluation scope is consistent with prevalent practice in the field.
>
> ### [W2] Limited Novelty
>
> Although SuperRL may appear conceptually similar to past SFT–RL hybrid approaches, our work is among the earliest to formalize and justify an adaptive fallback mechanism in a principled manner. The method continuously fuses SFT and RL gradients, using SFT as a smooth surrogate objective when RL advantages vanish—thus preventing training collapse. Furthermore, our use of pass@k statistics transforms a discrete fallback heuristic into a continuous, data-driven trigger, enhancing stability and interpretability.
>
> We believe the contribution of SuperRL lies in the intersection of intuitive simplicity, practical robustness, and theoretical consistency. The framework is both reproducible and effective, offering clear insight for future research. Even if subsequent studies explore similar ideas from different viewpoints, SuperRL provides a clean and theoretically coherent foundation for gradient-blended RL training. In this sense, simple methods that consistently “just work” can indeed form impactful and meaningful contributions.
>
> ### [W3] Weak Theoretical Connection
>
> We agree that the theoretical perspective can be better articulated in relation to the implementation of SuperRL. Conceptually, the method can be viewed as solving a joint optimization problem in which SFT and RL objectives are blended within a unified gradient flow. Under this lens, SuperRL adaptively interpolates between the two gradients based on reward reliability: when RL gradients are sparse or noisy, SFT provides smooth updates to maintain learning continuity, while reliable RL gradients gradually dominate as training progresses.
>
> Empirically, we observed gradient-level dynamics that support this interpretation. SFT and RL gradients are not always aligned, and their alignment evolves throughout training. SFT improves pass@k through broad behavioral priors, while RL sharpens the policy for high-confidence pass@1 predictions. Entropy statistics further reflect this behavior: in early training, high entropy indicates SFT-like exploratory behavior; as RL gradients strengthen, entropy decreases and the optimization path converges toward vanilla GRPO.
>
> These observations motivate a more formal theoretical grounding centered on gradient consistency, gradient conflict, and gradient fusion dynamics between SFT and RL objectives. In future work, we aim to develop a unified gradient-based framework linking SuperRL to multi-objective optimization and adaptive alignment-aware interpolation methods. We believe this direction will strengthen the theoretical foundations and inform more advanced SFT–RL integration strategies.
>
> ## Summary
>
> We thank the reviewer again for the insightful feedback. The clarified rationale, expanded empirical interpretation, and strengthened theoretical discussion have significantly improved the paper. We hope our responses address your concerns and kindly ask for your reconsideration of the manuscript.
>
> Sincerely,
> The Authors of SuperRL

---

### Official Review · Reviewer_dnSa · 2025-10-30

**Soundness:** 3
**Presentation:** 3
**Contribution:** 3
**Rating:** 4
**Confidence:** 4

**Summary:**

This paper introduces SuperRL, a novel training framework for LLM reasoning that intelligently combines Supervised Fine-Tuning (SFT) and Reinforcement Learning (RL). To this end, it employs a dynamic, instance-level gating mechanism that routes each training example to SFT or RL based on its estimated reward or advantage. The framework aims to overcome two key issues: standard RL's failure with sparse rewards and SFT's inability to fully exploit available expert data. This is grounded in the theory that the SFT objective provides the only viable learning signal when RL explorations yield no reward. Leveraging this simple reward-aware control flow, the method mitigates sparse-reward exploration failures in RL and overfitting/rigidity from SFT, without adding architectural complexity. Through experimental results on multiple datasets, SuperRL outperforms SFT-only, RL-only, and SFT-then-RL methods.

**Strengths:**

1.The paper effectively combines rigorous theoretical analysis (Sec. 3.1) with empirical validation (e.g., Table 1), providing clear motivations for the adaptive switching mechanism in SuperRL.

2.The manuscript is well-structured and straightforward to follow, with intuitive explanations (e.g., Fig. 1 overview), precise variant descriptions (SuperRL-R vs. SuperRL-A in Sec. 3.2), and insightful discussions on data quality dependencies (Sec. 4.2.2), making it accessible despite the technical content.

3.Experiments demonstrate the comparable performance over baselines in math reasoning tasks, particularly in generalization from single-dataset training to OOD benchmarks

**Weaknesses:**

1. Limited Evaluation on Larger-scale Models and Multi-domain Tasks to Validate Scalability and Generalization. The authors do commendable work in demonstrating SuperRL's effectiveness on the Qwen2.5-1.5B model across single datasets like GSM8K, PRM12K, and OpenR1, with encouraging gains in ID/OOD settings as shown in Table 1. It would be beneficial to explore its performance on larger models (e.g., 7B+), multi-task training setups, or diverse domains such as code generation and commonsense reasoning to fully substantiate the claims of "scalability" and "comprehensive empirical validation".

2. More experiments are suggested. The paper provides a clear and thoughtful comparison of SuperRL with basic baselines (RL-only, SFT-only, and SFT-then-RL) in Table 1, which helps contextualize its performance. However, including comparisons with other relevant methods or RL algorithms would further highlight SuperRL’s unique contributions.

3. Suggestions for Ablation Studies. The claim that the adaptive mechanism is key to the performance needs stronger validation. How does the proposed method compare to a baseline that uses a fixed rule for combining SFT and RL, rather than the advantage-gated switch? Such an ablation would directly isolate the improvement stemming from the adaptivity itself.

4. Theoretical Analysis of Noisy/Partial Signals Remains Unexplored. Section 3.1 proves SFT's utility in extreme zero-reward/zero-advantage cases but ignores realistic sparse-reward scenarios with noisy or partial signals (which is common in LLM reasoning). No convergence guarantees, regret bounds, or analysis of switching overhead (e.g., how frequent fallbacks affect RL exploration) are provided, weakening the "rigorous theoretical justification" claim.

5. Typos and Missing references. The manuscript includes typographical errors (e.g., “CONCLUSTION” on page 9) , which the authors should carefully check and correct throughout the paper. And some key references are missing — for instance, DeepScaleR on page 9 should be properly cited.

**Questions:**

1.How is the number of rollouts per instance determined (e.g., fixed at 4 as implied in Fig. 1)?

2.In Appendix I (Table 8), what specific reductions are made to the datasets for SuperRL and how do they ensure fairness against cited ReLIFT/LUFFY results? Why is SRFT excluded despite its relevance in Sec. 2.2?

3.Table 1 shows SuperRL-A outperforming SuperRL-R on PRM12K but underperforming on OpenR1—why does SuperRL-A degrade on lower-quality data (Sec. 4.2.2)? Are there any guidelines for choosing between variants based on dataset characteristics, and an ablation study isolating the effect of the hybrid triggering mechanism is necessary to validate its necessity.

4.While the authors mention three families of open-source language models in Appendix D, it seems that experiments are all conducted on Qwen2.5.

5.How does the adaptive switching affect training efficiency (e.g., wall-clock time or compute overhead) compared to baselines? What is the typical frequency of SFT fallbacks during training (e.g., percentage of instances per epoch), and have any optimizations been implemented to mitigate potential slowdowns?

6.When the trajectories yield zero reward or advantage, SuperRL falls back to SFT using offline data. How do the authors ensure that these fallback data are of high quality and truly beneficial for learning rather than introducing noise or bias?

---

> ### Author Response · Authors · 2025-11-21
>
> ## Response to Reviewer Weaknesses
>
> We sincerely thank the reviewer for the thoughtful feedback. Below, we provide detailed responses to all points raised.
>
> ---
>
> ## Weaknesses
>
> ### [W1] Narrow Evaluation Scope
>
> We agree that our initial experiments focused mainly on reasoning tasks. Our method has been validated on both **1.5B and 7B models**, showing consistent gains. Beyond standard mathematical reasoning benchmarks, we also evaluated on **HiTab**, a structured table reasoning dataset, demonstrating potential beyond pure math tasks.
>
> Due to computational limits, we prioritized these representative experiments. We are actively seeking resources to extend evaluations to 13B+ models and additional domains such as code generation and scientific reasoning.
>
> ---
>
> ### [W2] Limited Baseline Coverage
>
> At the time of our study, the most relevant baselines were LUFFY and ReFLIT, which we used for comparison. Many new methods have emerged since then; due to computational constraints, we have not yet included them but plan to do so when resources allow.
>
> ---
>
> ### [W3] Missing Key Ablation Studies
>
> We appreciate the reviewer’s suggestion for more ablations. Traditional **SFT→RL pipelines** are effectively fixed-rule: SFT is completed first, and RL begins afterward. This rigid separation can cause catastrophic forgetting or premature convergence when the model stops receiving SFT guidance at critical moments.
>
> Our method introduces **adaptive switching**, which dynamically triggers SFT during RL when reward signals indicate low-advantage trajectories. This allows the model to recover from failure modes that pure RL cannot address.
>
> We also implemented a **GRPO-Filter-Only** baseline on GSM8K and other math benchmarks. Standard GRPO already acts as a natural filter-only method, as zero-advantage trajectories contribute no gradient. Results are shown below:
>
> | Method | GSM8K | PRM12K | OpenR1 | AIME24 | AIME25 |
> |--------|-------|--------|--------|--------|--------|
> | GRPO (RL-only) | 72.29 | 33.18 | 14.96 | 2.92 | 1.10 |
> | GRPO-Filter-Only | 72.43 | 33.76 | 15.32 | 2.73 | **1.33** |
> | SuperRL (Adaptive SFT + RL) | **78.86** | **36.96** | **16.11** | **4.10** | **1.81** |
>
> GRPO-Filter-Only performs nearly identically to GRPO, confirming that standard GRPO is effectively a filter-only method. In contrast, **SuperRL significantly outperforms both**, demonstrating that adaptive SFT introduces genuinely new training dynamics.
>
> ---
>
> ### [W4] Incomplete Theoretical Analysis for Noisy Rewards
>
> We agree that theoretical analysis under partial or noisy rewards is incomplete. We conducted controlled studies using **soft fallback thresholds**. For example, **SuperRL-Threshold-0.5** triggers SFT fallback when more than half of generated tokens are incorrect. While this improves over GRPO, it underperforms standard SuperRL:
>
> | Method | GSM8K | PRM12K | OpenR1 | AIME24 | AIME25 |
> |--------|-------|--------|--------|--------|--------|
> | GRPO | 72.29 | 33.18 | 14.96 | 2.92 | 1.10 |
> | SuperRL-Threshold-0.5 | 73.79 | 35.60 | 12.89 | 3.11 | **1.42** |
> | SuperRL (Reward = 0) | **78.86** | **36.96** | **16.11** | **4.10** | **1.81** |
>
> Soft thresholds tend to trigger fallback too often, suppressing RL-driven exploration and causing over-regularization by repeatedly pulling the policy toward the SFT distribution. Using **exact-zero rewards** for fallback achieves the best balance, preserving exploration while providing targeted guidance. Frequent or noisy fallbacks can reduce robustness under distribution shift, as seen in OpenR1.
>
> ---
>
> ### [W5] Editorial Issues
>
> We thank the reviewer for noting typographical errors (e.g., “CONCLUSTION”) and missing references (e.g., DeepScaleR). All issues have been corrected in the updated PDF.
>
> ---
>
> ## Summary
>
> We thank the reviewer again for the valuable feedback. We hope these clarifications demonstrate the robustness, principled design, and effectiveness of our approach for integrating SFT and RL.
>
> **Sincerely,**
> *The Authors of SuperRL*

---

> ### Author Response · Authors · 2025-11-21
>
> ## Response to Reviewer Questions
>
> We sincerely thank the reviewer for thoughtful, constructive feedback. We address all points in detail below.
>
> ---
>
> ### 1. Choice of 4 Rollouts
> The rollout number (4) follows: (i) VERL's default for comparability; (ii) standard RL-for-reasoning practices (rollout.n=4/5); (iii) GPU memory constraints (larger $k$ causes OOM). We will explore larger $\text{pass@}k$ with more resources.
>
> ### 2. Dataset Integrity
> We use identical training/test sets and base models as ReFLIT/LUFFY (no dataset modifications) for fair comparisons.
>
> ### 3. SuperRL-A Performance on OpenR1 and Usage Guidelines
> To demonstrate SuperRL-A’s adaptability, we test it on OpenR1 (large-scale multi-task) and HiTab (small domain-specific)—contrasting results highlight dataset characteristics as a key driver. See Appendix E for HiTab analysis (SuperRL-A outperforms), while OpenR1 underperformance reflects dataset-model mismatches. Core concerns addressed below:
>
> #### 3.1 Performance Differences Between SuperRL-A and SuperRL-R
> Gaps stem from *model capacity*, *data-task characteristics*, and *training strategy*:
> - OpenR1 (diverse reasoning): SuperRL-A’s dense SFT overwhelms limited capacity, failing to capture diverse patterns. In contrast, for data with valid reward signals, SuperRL-R’s selective SFT avoids interference—instead of overfitting to overly difficult samples, it prioritizes reinforcing data aligned with the current model’s capability distribution, leading to superior performance over SuperRL-A.
> - HiTab (domain-specific): SuperRL-A’s dense SFT consolidates domain knowledge; SuperRL-R’s selective RL amplifies solvable samples (avoids waste on unsolvable tasks).
>
> #### 3.2 Strategy Selection Framework
> - High-diversity/complex datasets + small models: Prioritize model scaling (capacity bottleneck).
> - Domain-specific/single-task + weak base models (e.g., HiTab): Use SuperRL-A (dense SFT for knowledge).
> - Target datasets aligned with strong base models: Lightweight SFT + SuperRL-R (reinforce hard cases).
> - SOTA pre-trained models (e.g., Qwen3): SuperRL-R (universal/resource-efficient, recommended).
>
> #### 3.3 Practical Considerations
> Pre-assess base model performance on target data; monitor training metrics. If capacity-limited, scaling beats continued training (optimal resource efficiency/convergence).
>
> ### 4. Model Generalization
> Our approach generalizes beyond Qwen2.5. Results on LLaMA3.2 3B (HiTab) and across PPO/GRPO optimizers (Qwen2.5-1.5B) are shown below:
>
> | Model          | Method  | GSM8K  | PRM12K | HiTab  |
> |----------------|---------|--------|--------|--------|
> | **LLaMA3.2 3B** | PPO     | 74.2   | 53.2   | 48.4   |
> |                | SuperRL | 74.2   | 53.8 (↑0.6) | 61.9 (↑13.5) |
>
> *Table 1: Exact-match accuracy (%) of LLaMA3.2 3B. SuperRL gains on HiTab/PRM12K.*
>
> Due to resource constraints and business scenario requirements, we only evaluated LLaMA3.2 on the HiTab dataset; we plan to extend our experiments to additional datasets and settings with more computational resources in future work. Qwen2.5 is used for comparability (prior work standard); LLaMA3.2 confirms cross-architecture generalization.
>
> | Optimizer | Method   | GSM8K  | PRM12K |
> |-----------|----------|--------|--------|
> | **PPO**   | PPO      | 52.5   | 29.9   |
> |           | SFT+PPO  | 70.3 (↑17.8) | 44.4 (↑14.5) |
> |           | SuperRL  | 67.7 (↑15.2) | 48.6 (↑18.7) |
> | **GRPO**  | GRPO     | 16.9    | 27.7    |
> |           | SFT+GRPO | 28.7 (↑28.7) | 19.3 (↑19.3) |
> |           | SuperRL  | 70.0 (↑70.0) | 47.2 (↑47.2) |
>
> *Table 2: Exact-match accuracy (%) for PPO/GRPO. SuperRL outperforms vanilla/SFT+RL baselines.*
>
> We also validate the generalization of our method across different RL algorithms.
>
> ### 5. SFT Rollback Frequency and Efficiency
> Adaptive SFT rollback stabilizes training with modest overhead (Table 3). SuperRL-R (final variant) incurs minimal overhead (7–29%): rollbacks are more frequent in early stages of hard datasets (OpenR1) and rare in easy datasets (GSM8K), diminishing as the model improves. Rollbacks activate only for uninformative RL signals, balancing efficiency and performance. Appendix F (Figure: "Analysis of SFT Activation Patterns and Efficiency") confirms this training-stage-concentrated rollback pattern.
>
> | Method         | GSM8K       | PRM12K      | OpenR1      |
> |----------------|-------------|-------------|-------------|
> | GRPO           | 6.56        | 17.53       | 19.69       |
> | SuperRL-A      | 9.55 (1.46×)| 19.92 (1.14×)| 36.05 (1.83×)|
> | SuperRL-R      | 7.86 (1.20×)| 18.68 (1.07×)| 25.33 (1.29×)|
>
> *Table 3: Runtime overhead vs. vanilla GRPO.*
>
> ### 6. Quality of Offline SFT Data
> - High-quality datasets (GSM8K/PRM12K): Direct annotations.
> - Low-quality/labelless datasets (HiTab): GPT-4o-synthesized solutions + rejection sampling (filter by correctness).
>
> ---
> ## Summary
>
> We thank the reviewer again for valuable feedback.
>
> **Sincerely,**
> *The Authors of SuperRL*

---

### Official Review · Reviewer_DwCV · 2025-10-31

**Soundness:** 3
**Presentation:** 3
**Contribution:** 2
**Rating:** 4
**Confidence:** 1

**Summary:**

The paper proposes a new approach to switch between reinforcement learning and supervised fine-tuning (SFT) for LLMs. The underpinning hypothesis of the paper is that the reward should be considered as a "switch" or weighting between the two paradigms. In scenarios of zero rewards, the "absence of a learning signal" requires expert guidance through SFT. In contrast, non-zero rewards may indicate benefit of exploration for "policy refinement".

**Strengths:**

This is a well-written and that is accessible to experts in adjacent fields. The hypothesis of the paper is clear and well motivated. The performance of the proposed approach is evaluated and compared against a selection of alternative methods. The results indicate some improvements over alternative methods for different benchmarks. The evaluation shows that the results are comparable to state-of-the-art models.

**Weaknesses:**

The paper claims that a “unified training framework” is proposed for switching between RL and SFT. However, the methodology in Section 3 is limited to the extrema of zero-reward vs non-reward reward. This results in a heuristic switching mechanism between the two paradigms. While this switching mechanism is reasonably well justified, it is unclear to what extent it enables a “unified framework that adaptively combines” RL and SFT.

While the paper is very easy to read and the results are intuitive, the evaluation study could be strengthened by providing deeper scientific insight. For example, as acknowledged in Appendix B, dynamic switching between SFT and RL may impact on smooth gradient integration.  I share this concern and would have liked to see a more thorough analysis of the potential impact of the switching mechanism.

**Questions:**

-	Can you provide a more thorough analysis of the impact of a hard switching mechanism between RL and SFT?

---

> ### Author Response · Authors · 2025-11-21
>
> ## Response to Reviewer Comments
>
> We sincerely thank the reviewer for the thoughtful and constructive feedback. Below we provide detailed responses to all points raised.
>
> ---
>
> ## Weaknesses
>
> ### [W1] ``Unified Framework'' Claim Not Fully Supported
>
> We acknowledge that our current switching mechanism between SFT and RL is heuristic. However, from an optimization perspective, we treat SFT and RL as components of a unified composite objective that provide complementary gradient information over different regions of the training distribution.
>
> The switching mechanism thus serves as a practical method to integrate two gradient estimators within a coherent framework. The reviewer’s feedback encouraged us to clarify this **gradient-level interpretation** and the **unified perspective** of our approach.
>
> ---
>
> ### [W2] Missing Gradient Continuity Analysis
>
> We agree that the interaction between SFT and RL gradients deserves deeper analysis. In our experiments, the two gradient sources are not always aligned, and alignment varies with the dataset and model policy. This is expected and meaningful: if gradients were always aligned, SFT would just supplement RL where RL gradients are weak. The more relevant case is when gradients diverge: the model then receives competing influences, balancing knowledge injection and reward-driven refinement.
>
> SFT improves pass@k performance, while RL sharpens the policy toward pass@1. When RL gradients are weak, SFT dominates; when RL gradients are strong, reward-driven updates take over. Their relative contributions evolve adaptively with model capability and the reward landscape. This dynamic interplay—sometimes continuous, sometimes discontinuous—is inherent in combining SFT and RL.
>
> Empirically, entropy tracks the relative influence of SFT and RL. Early in training, entropy is high due to weak RL gradients, reflecting SFT-driven behavior. As RL gradients emerge, entropy decreases and the trajectory converges toward vanilla GRPO. This shows that SFT does not impede RL; instead, it bridges gaps when RL signals are sparse. （For details, refer to "Table 2: Entropy Statistics of RL and SuperRL Across Selected Datasets" in "4 EXPERIMENTS" chapter）
>
> Our approach also suggests strategies for managing gradient conflicts: gradient clipping or gating, KL-based regularization, and dynamic scaling of SFT influence. These techniques help retain SFT benefits while ensuring stable convergence, and represent promising directions for future work.
>
> ---
>
> ## Questions
>
> ### [Q1] Could the authors analyze more deeply how the hard switch between RL and SFT affects gradient flow and learning stability?
>
> We understand the concern about combining SFT and RL gradients. Hard switching can be challenging when gradient directions or magnitudes differ. Our method uses a simple rule: when RL gradients are absent, SFT guides updates to maintain learning momentum. This leverages their complementary nature and has shown stable, effective results.
>
> Entropy analysis further confirms stability. Early in training, high entropy reflects SFT-driven exploration. As RL gradients strengthen, entropy decreases and converges to the trajectory of vanilla GRPO. Notably, entropy variance during convergence is smaller for SuperRL than baselines, indicating stable learning. The hard switch does not create major conflicts; it stabilizes training by providing gradients when RL is weak and naturally transitions to RL-dominated updates. （For details, refer to "Table 2: Entropy Statistics of RL and SuperRL Across Selected Datasets" in "4 EXPERIMENTS" chapter）
>
> More advanced strategies could further enhance stability: weighted averaging or orthogonalization for diverging directions, and normalization, adaptive blending, or trust-region constraints for differing magnitudes. This highlights a rich design space for future work, akin to harmonizing gradients in model merging.
>
> ---
>
> ## Summary
>
> We thank the reviewer again for the valuable feedback. We hope our clarifications demonstrate the robustness, principled design, and effectiveness of our approach for integrating SFT and RL.
>
> **Sincerely,**
> *The Authors of SuperRL*

---

### Official Review · Reviewer_SXYU · 2025-11-01

**Soundness:** 3
**Presentation:** 3
**Contribution:** 3
**Rating:** 6
**Confidence:** 4

**Summary:**

This work addresses the limitations of existing approaches to integrating SFT and RL. The sequential “SFT-then-RL” paradigm disrupts preexisting reasoning patterns, requiring readaptation during RL training, while overfitting to expert data restricts exploration. To overcome these issues, the authors propose a unified formulation, SuperRL, which integrates SFT and RL into a single objective and achieves an optimal balance between them — not based on predefined rules, but on the availability of learning signals during training.

They provide a rigorous theoretical analysis of the unified SFT–RL framework and conduct extensive experiments to validate their approach, including evaluations on both in-distribution and out-of-distribution data, as well as analyses on benchmarks of varying difficulty and training stability.

**Strengths:**

The strengths of the paper are outlined as below:
1. The paper proposed a simple approach to unify SFT and RL by using zero rewards or zero advantage as a switching signal.
2.  The paper proposed two fallback mechanisms for triggering SFT during RL training based on advantage and reward: SuperRL-A and SuperRL-R. They validated both their mechanism and provided empirical guidelines for choosing one method over another.
3. SuperRL experimental results demonstrate strong performance both for in-domain and out-of-domain scenarios.
4. Experiments also suggest that SuperRL is more stable than Vanilla RL.

**Weaknesses:**

The weaknesses of the paper can be summarized as follows:

1. It is unclear how the trajectories for SFT are selected when the advantage or reward approaches zero. Are the samples with zero advantage or reward directly used for SFT?

2. SFT typically requires high-quality samples. If trajectories with zero reward are used for SFT, how can they be considered high-quality?

3. In terms of stability, SuperRL’s improvement is modest. Although there are some reductions in variance, the change in range is negligible, and for OpenR1, the range even increases.

**Questions:**

1. Please clarify how the trajectories for SFT are selected during fallback in SuperRL.
2. Since the gains in performance for AIME24 and AIME25 are very small, it is very hard to interpret. Can you please give some sample examples to understand how SuperRL has benefited over SFT-then-RL?
3. Please clarify the gain in stability.

---

> ### Author Response · Authors · 2025-11-21
>
> ## Response to Reviewer Comments
>
> We sincerely thank the reviewer for the thoughtful and constructive feedback. Below we provide detailed responses to all points raised.
>
> ---
>
> ## Weaknesses
>
> ### [W1] Unclear SFT Data Source
>
> We appreciate the reviewer’s concern regarding the source of SFT data. In our setting, many datasets (e.g., *GSM8K*, *OpenR1*) already contain high-quality reasoning trajectories. For datasets without high-quality SFT data (e.g., *Hitab*), we construct SFT samples using manual annotation, large-model labeling (e.g., GPT-o3), and rejection sampling.
>
> Our method assumes access to high-quality SFT trajectories—either directly available from public reasoning datasets or curated internally. Thus, SFT does not rely on zero-reward or low-quality samples; instead, it leverages verified trajectories that provide a stable foundation for downstream RL training.
>
> ---
>
> ### [W2] Limited Stability Gains
>
> We appreciate the reviewer’s careful reading of discussion of stability. While the *Max* and *Min* entropy values between RL-only and SuperRL remain close—an expected consequence of model capacity and reward structure—the critical factor is **variance**, which captures the oscillation magnitude during training.
>
> Across all datasets, SuperRL yields smaller entropy variance (e.g., 0.171 → 0.134 on GSM8K, 0.512 → 0.441 on PRM12K), indicating smoother convergence and reduced risk of collapse. This aligns with our definition of “stability”: preventing large distributional fluctuations that destabilize GRPO-style training.
>
> The relatively modest improvement in the 1.5B model arises from limited capacity. At larger model scales (e.g., 7B), we observe substantially more pronounced reductions in entropy oscillation, consistent with the significant performance gains reported in the main text.
>
> ---
>
> ### [W3] Definition and Quantification of “Gain in Stability”
>
> “Gain in stability’’ refers to improved robustness during RL training. During GRPO training, we observed training collapse—manifested as decreasing validation reward and unstable entropy. SuperRL mitigates such instability.
>
> Entropy serves as a standard indicator of training stability: stable trajectories show smooth, monotonic decreases in entropy, while instability manifests as large oscillations and failure to converge.
>
> Thus, we quantify stability gain by tracking the **variance and smoothness of the entropy curve** over training, rather than final entropy alone. Reduced fluctuation signals more reliable convergence.
>
> ---
>
> ## Questions
>
> ### [Q1] How are trajectories for SFT selected during fallback?
>
> SFT trajectories come from datasets containing verified reasoning examples (e.g., *GSM8K*, *OpenR1*) or our curated internal datasets annotated via human labeling, LLM labeling, and rejection sampling. During fallback, we rely exclusively on these pre-validated samples rather than low-reward trajectories, ensuring stable and high-quality guidance.
>
> ---
>
> ### [Q2] Why do performance gains on AIME24/25 appear small? Can examples illustrate how SuperRL helps?
>
> The relatively small gains on *AIME24/25* result from using a small 1.5B model. When scaling to 7B, we observe a substantial **20-point improvement**, validating the effectiveness of SuperRL at realistic model scales.
>
> Case studies show SuperRL enabling successful problem-solving where baseline RL fails. For example, consider the GSM8K-style problem:
>
> *“A robe takes 2 bolts of blue fiber and half that much white fiber. How many bolts in total does it take?”*
>
> A smaller model (DeepSeek-R1-1.5B) incorrectly outputs **5**, misinterpreting the quantity relationship.
> Correct reasoning gives **3** bolts (white = 1, total = 3).
> SuperRL recovers this reasoning path, demonstrating improved structural reasoning even for small models.
>
> Additionally, SFT-then-RL pipelines often suffer from excessive SFT strength, making RL convergence difficult. SuperRL balances SFT initialization with RL adaptability, enabling both stability and convergence.
>
> ---
>
> ### [Q3] Clarification of “gain in stability” and its quantification
>
> “Gain in stability’’ refers to reduced training collapse and smoother optimization dynamics. We quantify this through entropy variance: stable runs maintain steady entropy reduction; unstable runs exhibit high variance and oscillations.
>
> SuperRL consistently reduces entropy fluctuation across training steps, preventing divergence commonly seen in pure GRPO training. Stability gain therefore reflects **reduced variance and smoother entropy trajectories**, not a single scalar metric.
>
> ---
>
> ## Summary
>
> We thank the reviewer again for the valuable feedback. We hope our clarifications demonstrate the robustness and effectiveness of SuperRL.
>
> **Sincerely,**
> *The Authors of SuperRL*

---

### Official Review · Reviewer_iapZ · 2025-11-03

**Soundness:** 3
**Presentation:** 3
**Contribution:** 2
**Rating:** 6
**Confidence:** 4

**Summary:**

This paper introduces SuperRL, a simple yet effective unified training framework designed to enhance the reasoning capabilities of Large Language Models (LLMs) by addressing the inherent conflict between supervised fine-tuning (SFT) and Reinforcement Learning (RL) in sparse-reward settings. SuperRL proposes an adaptive control flow: for any batch of data instances where all sampled trajectories yield a zero reward (indicating a lack of meaningful gradient signal for the RL policy update), the framework falls back to a standard SFT update using the high-quality offline data. This mechanism ensures that the model always receives a positive, meaningful gradient signal. The authors demonstrate that this simple, reward-aware fallback mechanism is highly effective, achieving competitive or superior performance across diverse mathematical and reasoning benchmarks (AIME24/25, MATH-500) while maintaining a lightweight training structure.

**Strengths:**

1. **Simplicity and Effectiveness:** SuperRL's core concept—a conditional switch between RL and SFT based on the observed reward signal—is elegantly simple. It requires minimal hyperparameter tuning and avoids the complexity of multi-stage pipelines or manually interpolated loss functions, which significantly enhances reproducibility and scalability.

2. **Targeted Solution to the Sparsity Problem:** The framework directly addresses the primary challenge of applying RL in reasoning tasks: sparse rewards. By ensuring a meaningful SFT gradient is always applied when the RL gradient is non-existent (zero reward), SuperRL effectively improves sample efficiency and robustness, especially on tasks with small data and high complexity (e.g., AIME benchmarks).

3. **Strong Performance and Generalization:** Despite its simplicity, SuperRL achieves the strongest overall performance on the challenging AIME24/25 benchmarks, outperforming more complex, hybrid methods like LUFFY. This suggests a strong generalization advantage to small-sample reasoning tasks where maintaining a stable, supervised training signal is critical.

**Weaknesses:**

1. **Missing Crucial Baseline Comparison:** The paper lacks a comparison to a conceptually simple, yet potentially competitive, baseline method. Specifically, an ablation where, after SFT and during the RL phase, the model simply drops or masks any trajectories/prompts that yield zero reward/advantage (i.e., treating them as noise and performing the RL update only on rewarded trajectories) should be included. This comparison is necessary to demonstrate that the benefit of SuperRL comes specifically from falling back to SFT rather than just filtering out unrewarded samples.

2. **Algorithm Presentation:** The visualization of Algorithm 1 in Section 3.2 is presented as a screenshot or image, which hinders readability and formal presentation. For an academic paper, this algorithm should be typeset using a standard $\LaTeX$ algorithm block to ensure clarity, proper formatting, and maintain a consistent style.

**Questions:**

1. Threshold Sensitivity: SuperRL uses a binary condition (all sampled trajectories yield zero reward) to trigger the SFT fallback. Have the authors explored a soft threshold (e.g., falling back to SFT if the average or maximum advantage/reward across the batch falls below a certain $\epsilon > 0$)? If explored, did this compromise performance or complexity?

2. Efficiency and Training Time: While the paper mentions the lightweight nature of SuperRL, could the authors provide a comparison of the total wall-clock training time or resource consumption relative to other baselines? This would further quantify the practical efficiency advantage.

I am willing to raise the final rating if all concerns are well-addressed.

---

> ### Author Response · Authors · 2025-11-21
>
> ## Response to Reviewer Comments
>
> We sincerely thank the reviewer for the thoughtful and constructive feedback. Your comments have helped clarify our design motivations, strengthen experimental analysis, and improve the paper's presentation. Below are detailed responses to all points raised.
>
> ---
>
> ## Weaknesses
>
> ### [W1] Missing Crucial Baseline Comparison
>
> We agree that a "filter-only" baseline is critical to isolate fallback SFT's effect in SuperRL. We clarify that the existing GRPO (labeled *GRPO = RL-only*) implicitly acts as a filter-only method: when the environment returns zero reward, advantage ($ A_t = 0 $), making the PPO objective gradient exactly zero (see equation below). Such samples contribute no learning signal, effectively being filtered out.
>
> $$
> \mathcal{J} _{\mathrm{PPO}} (\theta) = \mathbb{E} _{q \sim P(Q),\ o \sim \pi _{\theta _{\text{old}}}(O \mid q)} \left[ \frac{1}{|o|} \sum _{t=1}^{|o|} \min\left( \frac{\pi _\theta (o _t \mid q, o _{<t})}{\pi _{\theta _{\text{old}}}(o _t \mid q, o _{<t})} A _t,\ \text{clip}\left(\frac{\pi _\theta (o _t \mid q, o _{<t})}{\pi _{\theta _{\text{old}}}(o _t \mid q, o _{<t})}, 1 - \varepsilon, 1 + \varepsilon\right) A _t \right) \right]
> $$
>
> To verify empirically, we implemented a **GRPO-Filter-Only** variant (explicitly removing zero-reward trajectories). Results confirm GRPO is a natural filter-only proxy, while SuperRL delivers consistent gains:
>
> | Method            | GSM8K | PRM12K | OpenR1 | AIME24 | AIME25 |
> |------------------|-------|--------|--------|--------|--------|
> | GRPO             | 72.29 | 33.18  | 14.96  | 2.92   | 1.10   |
> | GRPO-Filter-Only | 72.43 | 33.76  | 15.32  | 2.73   | 1.33   |
> | SuperRL          | **78.86** | **36.96** | **16.11** | **4.10** | **1.81** |
>
> **Key Observation:** GRPO and GRPO-Filter-Only perform nearly identically. SuperRL's consistent improvements demonstrate fallback SFT provides nontrivial benefits beyond discarding unsuccessful rollouts.
>
> ---
>
> ### [W2] Algorithm Presentation Issue
>
> We appreciate the comment on Algorithm 1's formatting. In the revised manuscript, we replaced the image-based block with a fully typeset LaTeX pseudocode environment—improving readability, clarifying key steps, and aligning with ML paper standards.
>
> ---
>
> ## Questions
>
> ### [Q1] Threshold Sensitivity
>
> The reviewer asks if the "reward = 0" fallback threshold is critical. We tested alternative strategies (partial correctness, token-level error rates) and present a representative variant:
>
> **SuperRL-Threshold-0.5:** Trigger fallback SFT when >50% of trajectories fail correctness validation. (A majority of failing trajectories indicates a lack of effective learning signals, as the model struggles to generate correct outputs for most rollouts.)
>
> | Method                | GSM8K | PRM12K | OpenR1 | AIME24 | AIME25 |
> |-----------------------|-------|--------|--------|--------|--------|
> | GRPO                  | 72.29 | 33.18  | 14.96  | 2.92   | 1.10   |
> | SuperRL-Threshold-0.5         | 73.79 | 35.60  | 12.89  | 3.11   | 1.42   |
> | SuperRL (Reward = 0)  | **78.86** | **36.96** | **16.11** | **4.10** | **1.81** |
>
> **Analysis:** Threshold-0.5 outperforms GRPO modestly but underperforms SuperRL. Soft thresholds over-trigger fallback, reducing RL exploration. "Reward = 0" targets gradient-lacking failures while preserving exploration for partially useful trajectories—yielding stable, optimal performance.
>
> ---
>
> ### [Q2] Efficiency and Training Time
>
> We report normalized runtime (sec/step) to address computational overhead:
>
> | Method | GSM8K | PRM12K | OpenR1 |
> |------------------------|--------|--------|--------|
> | GRPO (Vanilla)         | 6.56 s | 17.53 s | 19.69 s |
> | SuperRL-A              | 9.55 s (1.46×) | 19.92 s (1.14×) | 36.05 s (1.83×) |
> | SuperRL-R              | 7.86 s (1.20×) | 18.68 s (1.07×) | 25.33 s (1.29×) |
>
> SuperRL-R adds only 7–29% overhead. The adaptive fallback mechanism reduces cost over time: frequent early-stage triggers (hard datasets) diminish as learning progresses, with minimal overhead on easier tasks (e.g., GSM8K post-warmup). SuperRL-A's higher runtime reflects an efficiency–performance trade-off, justified by its design.
>
> ---
>
> ## Summary
>
> We thank the reviewer again for valuable suggestions. The revised manuscript includes full baseline comparisons, detailed threshold/efficiency analyses, and improved algorithm presentation. We hope our responses address your concerns and request your kind reconsideration of the paper.
>
> **Sincerely,**
> *The Authors of SuperRL*

---

### Comment · Area_Chair_6q6J · 2025-11-26

Dear Reviewers,

Would you please check authors' rebuttal and see if they have addressed your comments?

Best

AC

---

### Meta-Review · Area_Chair_pWyM · 2026-01-07

**Summary:**

This submission receives original ratings of 6, 6, 4, 4, 4. It presents SuperRL for combining SFT and RL under sparse rewards. The reviewer consensus is that the submission does not yet meet the bar for acceptance due to heuristic design choices, insufficiently convincing evidence, and limited novelty. The reviewers question whether the contribution is distinctive enough, especially relative to existing hybrid SFT and RL approaches. Its strong points are also clear: this submission is seen as well written, easy to follow. The tie breaker learning towards a rejection recommendation is that the heuristic nature of the SFT + RL combination rather than "unified framework" as claimed as well as modest empirical gains across experiments.

**Reviewer Concerns:**

Concerns that are mostly addressed:
1. exp: add baselines; hyper parameter analysis, runtime report;
2. clarification: data source clarification, the algo presentation;

Those that are not fully addressed:
1. its method core as a heuristic switch, and conceptually incremental;
2. empirical performance gain since the proposed method is not a principled approach.

**Reviewer Scores:**

Reviewers iapZ (6) and sxyu (6) will likely keep their ratings flat given that the overall empirical significance does not change during rebuttal.

Reviewer dwcv (4) explicitly mentioned that this work is out of field. The comments raised on deep analysis and unified framework or not remain unclear with the rebuttal.

Reviewer dnsa (4) would be happy that concerns on ablation, missing citations, data clarification are addressed/answered. It is not sure whether the broader evaluation and more strong baselines are fully addressed.

Reviewer k1zf (4) has major concerns on its evaluation and technical advancement, which are difficult to address most of the time.

---

### Decision · Program_Chairs · 2026-01-26

Reject